# Alternative Furrow Irrigation Combined with Topdressing Nitrogen at Jointing Help Yield Formation and Water Use of Winter Wheat under No-Till Ridge Furrow Planting System in Semi-Humid Drought-Prone Areas of China

**Jinzhi Wu [1], Haoyue Guan [1], Zhimin Wang [2], Youjun Li [1], Guozhan Fu [1], Ming Huang [1,\*] and Guoqiang Li [3,4,\*]**

[1] College of Agriculture, Henan University of Science and Technology, Luoyang 471003, China; yywujz@126.com (J.W.); ghy167x@126.com (H.G.); lyj@haust.edu.cn (Y.L.); guozhanfu@163.com (G.F.)
[2] College of Agronomy and Biotechnology, China Agricultural University, Beijing 100193, China; zhimin206@263.net
[3] Key Laboratory of Huang-Huai-Hai Smart Agricultural Technology, Ministry of Agriculture and Rural Affairs, Zhengzhou 450002, China
[4] Institute of Agricultural Economics and Information, Henan Academy of Agricultural Sciences, Zhengzhou 450002, China
\* Correspondence: huangming_2003@126.com (M.H.); gqli@hnagri.org.cn (G.L.)

**Abstract:** Benefiting from the high–farmland construction program in China, one–off irrigation can be guaranteed in most fields in semi–humid drought–prone areas in China. However, little information is available on water and nitrogen (N) management in wheat production under this condition. This study aimed to explore the effects of alternative furrow irrigation (AFI) and topdressing N fertilizer (TN) on wheat productivity under a no–till ridge–furrow planting system in semi–humid drought–prone areas. The experimental design was as follows: two furrow irrigation (FI) methods, namely, EFI (every furrow irrigation) and AFI (alternative furrow irrigation) with 75 mm at the jointing stage were set as the main treatments. Two topdressing N (TN) patterns, namely, NTN (0 kg ha$^{-1}$ of N) and TN (60 kg ha$^{-1}$ of N) along with irrigation were set as the secondary treatments. Moreover, a traditional planting practice with no irrigation and no topdressing N (NINTN) was set as control. In 2018–2020, a field experiment was carried out to investigate the effects on soil water, leaf chlorophyll relative content (SPAD) and net photosynthetic rate ($P$n), aboveground dry matter assimilates, grain yield, water use efficiency (WUE) and economic benefit. We found that both FI methods and TN patterns significantly influenced soil water content. Compared with NINTN, the soil water content in each combination of the FI method and TN pattern was effectively improved at the booting and anthesis stages, leading to the significant increase in SPAD and $P$n in leaves, post–anthesis dry matter accumulation (POA), grain yield, WUE and economic benefit of winter wheat. Compared with the EFI, averaged across years and TN patterns, the AFI technique increased the soil water storage at booting and anthesis stages and significantly improved the $P$n at early milk (4.9%) and early dough (7.5%) stages, POA (40.6%) and its contribution to grain (CRPOA, 27.6%), the grain yield (10.2%), WUE (9.1%) and economic benefit (9.1%). In addition, compared with the NTN, the TN pattern significantly increased the water computation by wheat from booting to maturity, enhanced leaf $P$n after anthesis and POA, and finally resulted in the increase in grain yield (14.7–21.9%) and WUE (9.6–21.1%). Thus, the greatest improvement in the leaf photosynthetic characteristics, aboveground dry matter assimilates, grain yield, WUE and economic benefit was achieved under AFITN treatment. Above all, it can be concluded that the AFITN with AFI of 75 mm and TN of 60 kg ha$^{-1}$ at jointing was an alternative management strategy for optimizing yield formation and water use of winter wheat. This study provided new insights into improving wheat productivity in drought–prone areas where one–off irrigation can be guaranteed.

**Keywords:** semi–humid drought–prone areas; winter wheat; alternative furrow irrigation; topdressing N; grain yield; WUE





## 1. Introduction

Wheat is one of the stable food crops and feeds about 30% of the world's population [1], and about 75% of all the wheat is produced from dryland including arid, semi–arid and semi–humid drought–prone areas [2]. In these areas, winter wheat (*Triticum aestivum* L.) is mainly planted, and rainfall is insufficient and does not coincide with the water demands during the winter wheat growing stage [3,4]. Consequently, winter wheat is frequently exposed to water stress during the growth period, which results in low and unsustainable wheat productivity [5–7]. Thus, it is important to develop a cultivation strategy that can efficiently alleviate water stress and improve water use efficiency (WUE) to facilitate higher crop productivity in drylands [4,8,9].

As one of the most effective strategies to alleviate water stress and increase WUE in dryland, irrigation can optimize the water supply under water stress and thus assure optimum crop growth, booting grain yield [7,10,11]. Traditionally, in most farmlands in drylands, irrigation is unattainable for winter wheat due to the insufficient water resource and backward facilities for irrigation manufacturing, which has rendered the increase in wheat yield and the development of an irrigation strategy. Fortunately, in recent years, the situation is being improved with the plan for China's high–standard farmland construction; 66.7 million ha of high–standard farmland was constructed by the end of 2022, and 13.3 million ha are in plan during the 14th 5-year plan (2021–2025) [12]. This will improve the irrigation condition outstandingly and bring a chance to optimize irrigation management and boost winter wheat yield. In fact, a deficiency in agricultural water in these newly constructed farmlands still exists due to the limited water resource in dryland. Usually, there is only one opportunity for irrigation (one–off irrigation) during the wheat growing period [12]. However, the effects of one–off irrigation on crop productivity and coping management strategies are still limited. These situations have forced some farmers to adopt no irrigation during the wheat growth stage as usual, and some farmers tend to apply too much water with flood irrigation method to the wheat crop when available, which has aggravated the regional water deficiency and also affected the wheat growth negatively [13]. Therefore, it is necessary to a develop one–off irrigation strategy and evaluate its effects to increase the grain yield and water productivity of winter wheat.

Furrow irrigation (FI), as one of the important irrigation methods in which water moves in–furrow and crop is planted in–furrow, has been demonstrated as a potent tool for irrigating many field crops that are planted in rows [14–17]. It has many advantages such as water savings, low cost, greater simplicity, easy implementation and convenience for household use. More importantly, it can increase crop yield and water productivity [7,18,19]. Studies have shown that in the case of a proper design of furrow irrigation methods, water resources can be used more effectively [20,21]. The improved alternative furrow irrigation (AFI) not only has the advantages of the traditional every furrow irrigation (EFI) method but also has the advantages of controlling transpiration and reducing plant redundancy growth [11,21,22]. Previous studies have demonstrated that AFI significantly improved the soil water and nutrient distribution and thus promoted crop yield and water productivity [22]. Therefore, AFI is better applied to crop production, particularly in water–restricted areas, compared with EFI [19,20,23]. Topdressing fertilizers, especially topdressing N (TN), were an important strategy to increase crop WUE by providing nutrition for the following growing stages and boost crop productivity [24,25]. Considering the findings of Ji et al. [25], TN at jointing could help wheat meet the N demand and improve grain yield and N use traits. In fact, in traditional planting methods in drylands and semi–humid drought–prone areas, farmers prefer to apply all of the N fertilizer at once before sowing winter wheat, owing to the lack of irrigation and labor. This method results in an N supply shortage at the later growth stage, thereby reducing the water use and yield formation in winter wheat. Therefore, AFI and TN techniques are critical for establishing an efficient water use strategy that improves winter wheat yield and water productivity in dryland.

Ridge–furrow planting (RF), with two components comprising ridges (rainfall harvesting zone) and furrows (planting zone), is one of the important water–saving agricultural strategies. RF has been widely applied to improve wheat yield and WUE in dryland [11]. The RF system made irrigation easy to implement and convenient, retain water within the root zone in the furrow and increase wheat yield and WUE [11,18]. However, the wheat yield under the RF system without irrigation mainly relying on rainfalls is lower and unstable [26]. Previous studies have demonstrated that supplemental irrigation significantly increased the winter wheat yield under RF in dryland [9,11]. For example, Li et al. [9] found that a total amount of irrigation of 165 mm during the growth stage increased winter wheat yield by 46.1% in a dry year. Luo et al. [13] reported that the irrigation of 7.8–11.8 mm at wintering and jointing based on soil water measurements increased winter wheat yield by 10.0–27.1%. However, the previous literature mainly focused on the RF system with plastic films, which has resulted in serious soil and surface contamination caused by plastic sheets [27,28]. Recently, a novel RF system without plastic film named NTRFS has been widely used in the semi–arid and semi–humid drought–prone areas of China. NTRFS is a type of RF system with two rows of wheat planted in the furrow by a no–till fertilizer seeder [29]. Compared with the traditional flat planting practice, the NTRFS system significantly increased the grain yield and WUE in winter wheat and was widely used in farmers' fields [30]. However, the corresponding managements such as irrigation and N fertilizer application for the NTRFS system are still obscure, which limited the improvement of winter wheat productivity. Thus, it is of great importance and significance to study the effects of AFI and TN on the NTRFS system, especially in semi–humid drought–prone areas where the water supply has been improved with the high–standard farmland construction but still cannot meet sufficient irrigation for crop production in China and worldwide.

Above all, determining the effects of FI, AFI, TN and their combinations on soil water, leaf photosynthetic characteristics, dry matter accumulation and translocation, grain yield and water productivity will provide insights for the development of water and N management in dryland farming systems, which rely on limited irrigation. In the present study, AFI and TN techniques were used at the jointing stage of winter wheat in semi–humid drought–prone areas, where most fields have already had the chance of one–off irrigation, owing to the high farmland construction of China. The objectives were as follows: (1) to investigate the effects of FI, AFI, TN and their combinations on soil water, leaf SPAD and $Pn$, dry matter accumulation and translocation, grain yield and WUE; (2) to assess the contribution of different factors (FI, AFI, TN and interaction between AFI and TN) to the yield and WUE in winter wheat; and (3) to identify an optimized adaptive agronomic management strategy based on the synergistic effect of grain yield and water use for winter wheat under the NTRFS system in semi–humid drought–prone areas.

## 2. Materials and Methods

### 2.1. Study Site Description

From October 2018 to June 2020, the two–year field experiments were conducted at Nandasu village (34.82 N, 112.36 E) in Xiaolangdi town of Mengjin district of Luoyang city in Henan province, which is a typical dryland winter wheat production area in the semi–humid drought–prone areas of China. The average local annual air temperature is 13.7 °C, the mean annual frost–free period is 210 days, the average annual amount of sunshine is 2196 h, and the average annual precipitation is 650 mm. Approximately 60% of the annual precipitation occurs between June and September, which is just out of line with the wheat growth period. The winter wheat summer maize is practiced as the main cropping system, with the winter wheat planted in early or middle October and harvested in early June of the following year. The precipitation levels at the experimental site in the wheat growth season were 99.6 and 263.1 mm, respectively, accounting for 16.5% and 38.0% of the annual precipitations in 2018–2019 and 2019–2020 (Figure 1). Soils of the experimental site were developed on the cinnamon parent material and classified as calcareous Eum–orthic

Anthrosol (Chinese soil taxonomy), and the main properties in the 0–20 cm and 20–40 cm soil layers at the experiment initiation are listed in Table 1.

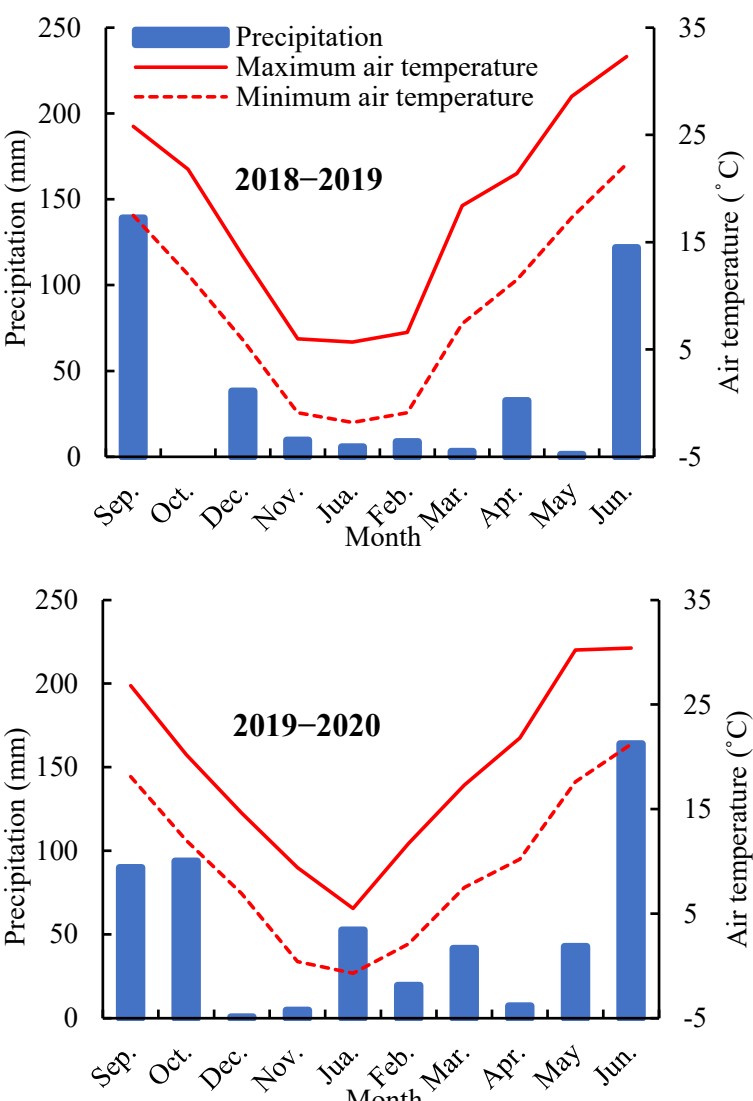

**Figure 1.** Monthly precipitation and maximum/minimum air temperature in 2018–2019 and 2019–2020 in the experimental field.

**Table 1.** Properties in the 0–20 cm and 20–40 cm soil layers sampled in the experimental field at sowing of winter wheat in 2018–2019 and 2019–2020.

| Year | Soil Depth (cm) | FC (%) | BD (g cm$^{-3}$) | SOM (g kg$^{-1}$) | TN (g kg$^{-1}$) | AP (mg kg$^{-1}$) | AK (mg kg$^{-1}$) | pH (H$_2$O) |
|---|---|---|---|---|---|---|---|---|
| 2018–2019 | 0–20 | 27.4 | 1.36 | 13.2 | 0.81 | 13.2 | 125.4 | 8.2 |
| | 20–40 | 26.8 | 1.41 | 11.3 | 0.74 | 10.1 | 110.9 | 8.4 |
| 2019–2020 | 0–20 | 27.3 | 1.35 | 13.1 | 0.81 | 12.1 | 121.6 | 8.2 |
| | 20–40 | 25.9 | 1.42 | 11.2 | 0.75 | 10.6 | 113.5 | 8.4 |

Note: FC = field capacity, determined by cutting ring method; BD = soil bulk density, determined by cutting ring method; SOM = soil organic matter, determined by dichromate wet oxidation; TN = total nitrogen, determined by Kjeldahl distillation–titration; AP = available phosphorus, determined by molybdenum–blue colorimetry; AK = available potassium, determined by ammonium acetate extraction/flame photometry. The pH was determined using a pH meter (Lei–Ci PHSJ–4F, Shanghai Instruments, China) with a 1:2.5 soil: water ratio (*w/v*). The determination methods were according to Bao [31].

### 2.2. Experimental Design and Field Management

We applied a two–factor experimental treatment in addition to the control. The treatments comprised two irrigation methods: (i) every furrow irrigation (EFI) and (ii) alternative furrow irrigation (AFI) (Figure 2) with two topdressing N patterns, (i) 0 kg N ha$^{-1}$ (NTN) and (ii) 60 kg N ha$^{-1}$ (TN), at the jointing of winter wheat. A traditional planting system of no irrigation with no topdressing N (NINTN) was used as the control. Thus, the experiment comprised a total of five treatments, e.g., no irrigation with no topdressing N (NINTN), every furrow irrigation with no topdressing N (EFINTN), alternative furrow irrigation with no topdressing N (AFINTN), every furrow irrigation with topdressing N (EFITN) and every furrow irrigation with topdressing N (AFITN). The amount of irrigation and fertilizer application rates were recommended by local agricultural experts and are listed in Table 2.

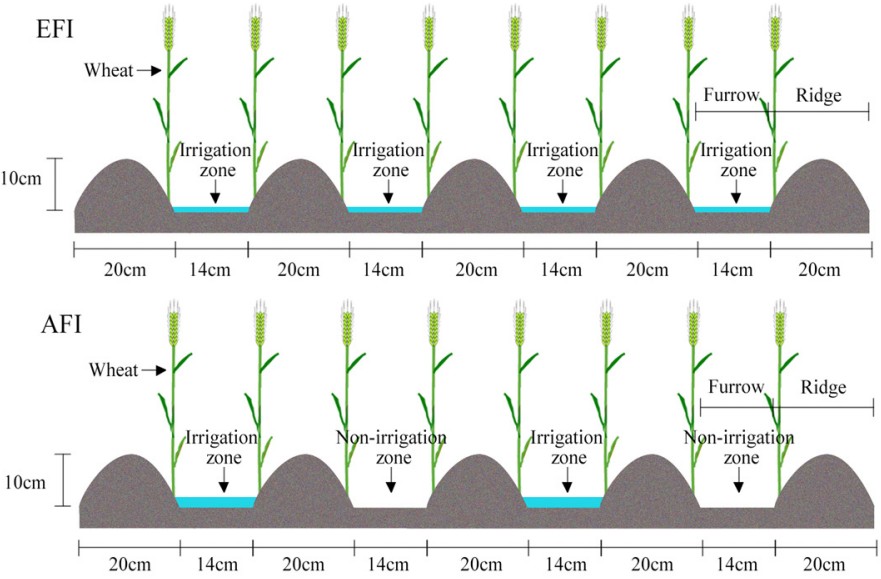

**Figure 2.** A schematic diagram of the field layout of EFI and AFI method.

**Table 2.** The irrigation amount and fertilizer application rates in different treatments in 2018–2019 and 2019–2020.

| Treatments | Irrigation (mm) | Fertilizer Application Rates (kg ha$^{-1}$) | | | |
|---|---|---|---|---|---|
| | | Basal N | Basal P$_2$O$_5$ | Basal K$_2$O | Topdressing N |
| NINTN (control) | 0 | 172.5 | 75 | 45 | 0 |
| EFINTN | 75 | 172.5 | 75 | 45 | 0 |
| AFINTN | 75 | 172.5 | 75 | 45 | 0 |
| EFITN | 75 | 172.5 | 75 | 45 | 60 |
| AFITN | 75 | 172.5 | 75 | 45 | 60 |

A completely randomized plot design with three replicates (plot size = 20 m × 6.12 m, all the replicates in the same field) was applied for all treatments, and the interval between plots was 1 m. At sowing, the compound fertilizers (N:P$_2$O$_5$:K$_2$O = 23:10:6) at the rate of 750 kg ha$^{-1}$ were applied as basal using a no–tillage fertilizer seeder (2BMQF–6/12A, Luoyang Xinle Machinery Co., Ltd., Luoyang, China). The 2BMQF–6/12A seeder can simultaneously conduct the operation of furrowing, ridging, fertilizing, sowing and re-pressing. The winter wheat cultivar Zhoumai36 at the seeding rate of 187.5 kg ha$^{-1}$ was drill–planted in the furrows with a space of 14 cm, and the basal fertilizers were drill–placed middle of the two seed rows at a depth of 10 cm. After sowing, the ridges and furrows were formed in the field (Figure 2). The width and height of the ridge were 20 cm and 10 cm, respectively, and the width of the furrow was 14 cm. Thus, the space of the wheat in

the wide row was 20 cm while that in the narrow row was 14 cm, with an average space of 17 cm. The winter wheat was sown on 13 October 2018 and 15 October 2019 and harvested on 30 May 2019 and 2 June 2020, respectively. Irrigation and topdressing N were carried out at the jointing stage (Zadoks 31) on 19 March 2019 and 22 March 2020, respectively. The irrigation amount was 75 mm both in AFI and EFI treatments, which was calculated based on the whole area of the plot and controlled by mechanical water meter reading (accuracy of 0.01 $m^3$, and the working pressure of the outlet valve was 0.10–0.12 MPa). In detail, the irrigation volume in each plot was 9.18 $m^3$, and the irrigation area in one furrow was 2.8 $m^2$ (20 m × 0.14 m). Under EFI treatments, the total irrigation area in each plot was 50.4 $m^2$ (20 m × 2.52 m), and the irrigation volume in one irrigated furrow was 0.51 $m^3$. Under AFI treatments, the irrigation area in each plot was 25.2 $m^2$ (20 m × 1.26 m), and the irrigation volume in one irrigated furrow was 1.02 $m^3$. For an irrigation furrow, the irrigation amount under AFI was 2–fold higher than that under EFI (Figure 2). For the TN pattern, the N amount of 60 kg $ha^{-1}$ was evenly broadcasted by hand in irrigation furrows just before irrigation. Weeds, pests and diseases were controlled with herbicides and pesticides by local farmers.

### 2.3. Measurements and Methods

### 2.3.1. Soil Water

The soil gravimetric water content (SWC) was determined periodically at a soil depth of 0–200 cm with intervals of 20 cm at sowing from the experimental filed, as well as at booting (Zadoks 43), anthesis (Zadoks 65) and maturity (Zadoks 94) during 2018–2019 and 2019–2020 years. From each plot, three or six random core samples were collected using a hand–held soil ferric auger (inner diameter = 4.0 cm), and the values were averaged. For the NI and EFI treatments, the soil cores were sampled from the middle of two rows in random furrows. For the AFI treatments, three soil cores were collected from the middle of the irrigated furrows, and another three cores were collected from the middle of the non–irrigated furrows. The soil samples from the same layer in the same plot within the same irrigation furrows were merged, and about 300 g of thoroughly mixed soil was collected and sealed immediately in a marked plastic bag for subsequent analysis. The soil water content was determined gravimetrically by drying in an oven at 105 °C for 24 h.

Soil water storage (SWS, mm) was calculated using the following equation [12]:

$$SWS = \sum_{i=1}^{n} D_i \times H_i \times W_i \times 10 \div 100$$

where $D_i$ is the soil bulk density (g $cm^{-3}$); $H_i$ is the soil thickness of the i layer (cm); $W_i$ is soil water content on a gravimetric basis (%); and n is the number of soil layers; i = 20, 40, 60, . . . , 200.

Water consumption (WC, mm) from booting to anthesis ($WC_{ba}$) and from anthesis to maturity ($WC_{am}$) was calculated as [4]:

$$WC = \Delta SWS + P$$

where $\Delta SWS$ (mm) is the difference in SWS between the beginning and the end of the period and P (mm) is the precipitation during the wheat growth period.

Evapotranspiration (ET, mm) over the whole winter wheat growing season was calculated as follows [12]:

$$ET = SWSs + P + U - R - F - SWS_m$$

where SWSs and $SWS_m$ are the SWS in the 0–200 cm soil layer at sowing and maturity, respectively; P (mm) is the precipitation during the wheat growth period; and I (mm) is the irrigation account. In our experiment, because the plots were surrounded by ridges to prevent the runoff of rainfall and irrigation water and the groundwater is buried deeper than 10 m in the soil layer, R and F are zero.

### 2.3.2. Leaf SPAD and Net Photosynthesis Rate (Pn)

At the booting (Zadoks 43), anthesis (Zadoks 65), early milk (Zadoks 73) and early dough (Zadoks 83) stages, the chlorophyll relative content (SPAD) of wheat leaves (at the same leaf position avoiding major veins) was recorded using a hand–held dual–wavelength chlorophyll meter (SPAD 502, Minolta Camera Co., Ltd., Osaka, Japan). Consistent with the SPAD measurements, the leaf net photosynthesis rate (*Pn*) was measured using an LI–Cor LI–6400 XT Portable Photosynthesis System (LI–COR Biosciences, Licoln, NE, USA) equipped with a LED leaf chamber. Measurements were taken on sunny days in the morning (9:00–11:00 a.m.) to avoid potential stomatal closure during the middle of the day. Each time, ten leaves of the center rows of each plot were measured at the same leaf position avoiding major veins, of which ten leaves for NINTN, EFINTN and EFITN treatment, five leaves of the irrigated row and five leaves of the no–irrigated row in each plot for AFINTN and EFITN treatments were measured. Before the flag leaf stage, the fully expanded top leaves were measured, and after that period, flag leaves were measured.

### 2.3.3. Aboveground Dry Matter Accumulation

At the anthesis (Zadoks 65) and maturity (Zadoks 94) stages, the four 0.5 m long wheat samples were cut from different rows in each plot. After cutting off the roots with scissors at the point where the stem met the roots, samples were separated into three components (stem plus sheath plus leaf and rachis plus glume) at anthesis and four components (stem plus sheath plus leaf, rachis plus glume and grain) at maturity. The aboveground parts were oven dried at 105 °C for 30 min and then at 65 °C for 24 h to determine the water content.

The aboveground dry matter accumulation (kg ha$^{-1}$) was calculated from the summed dry matter accumulation by each organ.

Pre–anthesis dry matter translocation (PRT, kg ha$^{-1}$) = dry matter accumulation at anthesis—dry matter accumulation in vegetative organ at anthesis.

Post–anthesis dry matter accumulation (POA, kg ha$^{-1}$) = dry matter accumulation at maturity—dry matter accumulation at anthesis.

Contribution rate of post–anthesis dry matter accumulation to grain yield (CRPOA, %) = post–anthesis dry matter accumulation ÷ dry matter accumulation in grain at maturity × 100.

### 2.3.4. Grain Yield, Yield Components and Harvest Index (HI)

At maturity, four 2 m × 1.36 m sampling areas were selected randomly in each plot, and the plants were harvested manually to determine the grain yield. After air drying, the sampled plants were threshed and the grain was weighed, where subsamples of 100 g of air–dried grain were oven dried at 90 °C for 30 min and then at 65 °C for 24 h to determine the grain water content and dry weight. The grain yield for each plot was expressed at a moisture content of 12.5% and calculated according to the air–dried grain weight and its water content. Meanwhile, 100 plants were randomly sampled to determine grains per spike and the 1000–grain weight. HI was determined as the ratio of the grain yield relative to the biomass yield.

### 2.3.5. Water Use Efficiency

Water use efficiency (WUE, kg ha$^{-1}$ mm$^{-1}$) was calculated using the following equations [12]:

$$WUE = Y \div ET$$

where Y (kg ha$^{-1}$) is the grain yield and ET (mm) is the evapotranspiration over the whole winter wheat growing season.

### 2.3.6. Economic Benefit

Economic benefit (USD ha$^{-1}$) was calculated as the difference between grain income and the input value. The input value consisted of the cost of materials (including wheat seeds, fertilizer, herbicide, pesticide and irrigational water), labor (including herbicide and

pesticide spraying and irrigating) and machinery application (including one–off operation of furrowing, ridging, fertilizing, sowing and repressing and harvesting). All costs were fixed because the materials, labor and machinery application were the same in the two growing seasons. The income refers to the value of the wheat grain.

$$\text{Benefits cost rate} = \text{Benefits}/\text{Inputs}$$

### 2.3.7. Contributions of Different Factors to the Grain Yield and WUE

We studied the quantitative effects of three factors comprising the FI, AFI, TN and the interaction between AFI and TN on the grain yield and WUE, where the NINTN treatment was the baseline. The contributions of these three factors to the wheat grain yield and WUE were calculated according to Zhang et al. [4], as shown in Table 3.

**Table 3.** The calculated equation of contributions of different factors to the grain yield and WUE.

| Index | Level | Calculated Equation |
|---|---|---|
| $C_0$ | – | $C_0$ = grain yield (WUE) under NINTN |
| $C_{fi}$ | NTN | $C_{fi}$ = 1/2 grain yield (WUE) under EFINTN + 1/2 grain yield (WUE) under AFINTN − grain yield (WUE) under NINTN |
| | EFI | $C_{fi}$ = grain yield (WUE) under EFINTN − grain yield (WUE) under NINTN |
| | AFI | $C_{fi}$ = grain yield (WUE) under AFINTN − grain yield (WUE) under NINTN |
| $C_{afi}$ | NTN | $C_{afi}$ = grain yield (WUE) under AFINTN − grain yield (WUE) under EFINTN |
| | TN | $C_{afi}$ = grain yield (WUE) under AFITN − grain yield (WUE) under EFITN |
| $C_{tn}$ | EFI | $C_{tn}$ = grain yield (WUE) under EFITN − grain yield (WUE) under EFINTN |
| | AFI | $C_{tn}$ = grain yield (WUE) under AFITN − grain yield (WUE) under AFINTN |
| $C_{at}$ | – | $C_{at}$ = (1/2 grain yield (WUE) under AFINTN – 1/2 grain yield (WUE) under EFINTN + 1/2 grain yield (WUE) under AFITN − 1/2 grain yield (WUE) under EFITN) − (1/2 grain yield (WUE) under EFITN − 1/2 grain yield (WUE) under EFINTN + 1/2 grain yield (WUE) under AFITN − 1/2 grain yield (WUE) under AFINTN) |

Note: $C_0$, Baseline; $C_{fi}$, Contribution of FI factor to yield (WUE); $C_{afi}$, Contribution of AFI factor to yield (WUE); $C_{tn}$, Contribution of TN factor to yield (WUE); $C_{at}$, Contribution of AFI and TN factor interaction to yield (WUE).

### 2.4. Statistical Analysis

Data processing and analysis of variance (ANOVA) were conducted using SPSS 18.0. Means of the data for each treatment were calculated by averaging the values for each plot. The least significant difference (LSD) test was used to detect differences among means among the treatments at a 5% probability level. The graphs were prepared using Microsoft Excel 2010.

## 3. Results

### 3.1. Soil Water

There were obvious differences in soil water storage (SWS) among the treatments at booting, anthesis and maturity stages in both years (Table 4, Figure 3). Compared with the NINTN treatment, the FI treatment significantly improved the SWS by 8.7–12.9% and 5.2–8.2% at the booting and anthesis stages, respectively, but the SWS was decreased by 1.8% (2018–2019, $p > 0.05$) −3.2% (2019–2020) at the maturity stage under TN. Compared with EFI treatment, AFI treatment significantly decreased the SWS by 4.1% in the 0–40 cm soil layer at the booting stage, while the SWS was increased by 5.2% and 3.0% at 80–160 cm at the booting and anthesis stages, respectively. This indicated that AFI promoted irrigation water to infiltrate into the subsoil layer. Compared with NTN treatment, TN treatment did not affect the SWS at the booting and anthesis stages but increased the absorption of soil water by wheat, thus decreasing the SWS by 1.7–3.8% at the maturity stage, especially the SWS at the maturity stage in the 60–160 cm soil layer, which decreased significantly. In the 2019–2020 year, due to the higher precipitation from January to March, the SWS was higher

than that in the 2018–2019 year, and the effect of AFI on SWS in subsoil was enhanced more at the booting and anthesis stages.

### 3.2. Leaf SPAD and Pn

Significant differences in SPAD and $P$n were noticed among the five treatments in both years (Figure 4). Compared with NINTN treatment, all four FI treatments significantly improved the leaf SPAD by averages of 4.9–11.4%, 9.7–12.7%, 8.6–22.3% and 14.7–25.6% at booting, anthesis, early milk and early dough stages, as well as $P$n by averages of 6.0–14.0%, 5.1–13.6%, 15.7–29.4% and 26.4–55.6%, respectively, averaged across the two years. Compared with the EFI treatment within the same TN pattern, AFI treatment decreased SPAD and $P$n in leaves a little at the booting stage but increased SPAD and $P$n at anthesis and significantly increased SPAD by 3.8–6.1% and 7.3–8.3%, as well as $P$n by 4.1–5.4% and 3.0–8.4%, at early milk and early dough stages, respectively, averaged across the two years. Compared with NTN treatment within the same FI method, TN treatment significantly improved the leaf SPAD by 4.7%, 3.2%, 9.4% and 10.7%, as well as $P$n by 6.2%, 5.6%, 6.7% and 15.0%, respectively, at booting, anthesis, early milk and early dough stages, averaged across the years and FI methods. Thus, the average leaf SPAD and $P$n were ranked as AFITN > EFITN > AFINTN > EFINTN > NINTN with a significant difference in most conditions after anthesis.

**Table 4.** Soil water storage (SWS) at booting, anthesis and maturity stages affected by different treatments in the 2018–2019 and 2019–2020 years.

| Treatment | Booting Stage | | Anthesis Stage | | Maturity Stage | |
|---|---|---|---|---|---|---|
| | 2018–2019 | 2019–2020 | 2018–2019 | 2019–2020 | 2018–2019 | 2019–2020 |
| NINTN | 412 c | 428 c | 383 c | 403 c | 341 b | 370 a |
| EFINTN | 452 b | 475 b | 406 b | 430 ab | 349 a | 372 a |
| AFINTN | 461 a | 481 a | 414 a | 436 a | 345 ab | 372 a |
| EFITN | 448 b | 476 b | 404 b | 424 b | 343 b | 358 b |
| AFITN | 460 a | 483 a | 408 ab | 430 ab | 335 c | 359 b |
| | *F*-value | | | | | |
| Year (Y) | 736.9 ** | | 975.2 ** | | 2146.7 ** | |
| Treatment (T) | 380.2 ** | | 149.8 ** | | 85.7 ** | |
| Y × T | 4.2 * | | 1.6 ns | | 10.7 ** | |

Note: NINTN, traditional no irrigation with no topdressing N; EFINTN, every furrow irrigation with no topdressing N; AFINTN, alternative furrow irrigation with no topdressing N; EFITN, every furrow irrigation with topdressing N; AFITN, alternative furrow irrigation with topdressing N. Means in a column followed by the different lowercase letters within a year are significantly different at $p < 0.05$. * Significant at $p < 0.05$; ** significant at $p < 0.01$; ns, not significant.

### 3.3. Aboveground Dry Matter Accumulation and Translocation

Although a decrease in dry matter accumulation at anthesis (DAA) and its translocation to grain (PRT) under AFI (TN) treatments was found compared with EFI (NTN), the post–anthesis dry matter accumulation (POA) and its contribution to grain (CRPOA) were improved significantly under FI, AFI and TN in both years (Table 5). Compared with EFI treatment within the same TN pattern, AFI treatment increased the aboveground dry matter accumulation at maturity (DAM) with averages of 7.1%, as well as DAA, PRT and POA by 40.6%, 27.6% and 36.3%, respectively, averaged across years and TN patterns. The increase was especially greater under the NTN pattern. Compared with NTN treatment within the same FI method, TN treatment significantly improved the DAA and DAM by 3.0% and 12.9%, as well as the POA and CRPOA by 46.4% and 23.7%, respectively, averaged across the years and FI methods. The greatest DAM was achieved under the AFITN treatment, which was 53.4%, 20.8%, 9.8% and 4.1% higher than that under NINTN, EFINTN, AFINTN and EFITN treatments, respectively.

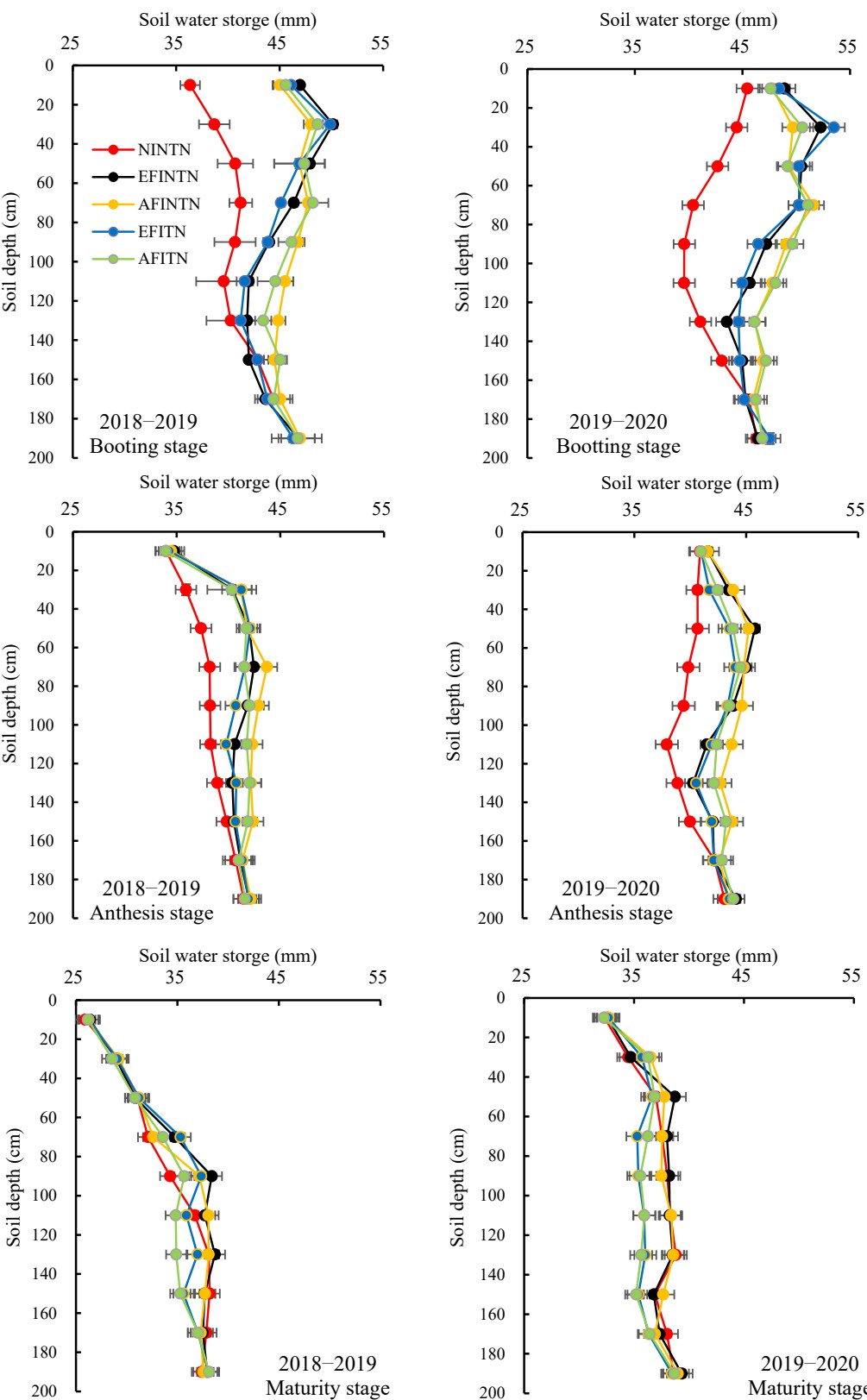

**Figure 3.** Soil water storage (SWS) in the different soil layers at booting, anthesis and maturity stages affected by different treatments in the 2018–2019 and 2019–2020 years. NINTN, traditional no irrigation with no topdressing N; EFINTN, every furrow irrigation with no topdressing N; AFINTN, alternative furrow irrigation with no topdressing N; EFITN, alternative furrow irrigation with topdressing N; AFITN, alternative furrow irrigation with topdressing N. Bars indicated standard deviation.

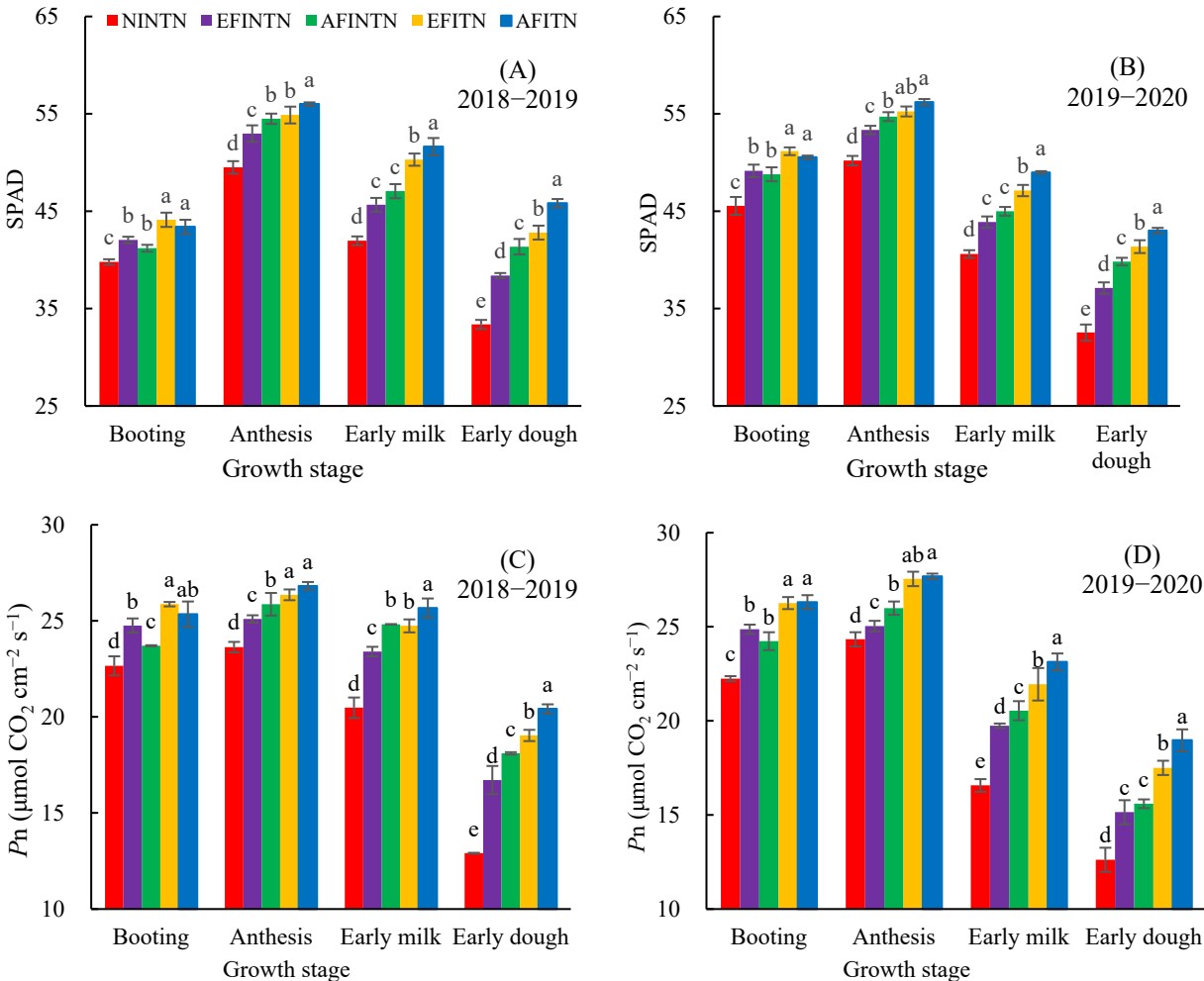

**Figure 4.** Leaf SPAD (**A**,**B**) and *P*n (**C**,**D**) at the different growth stages affected by different treatments in the 2018–2019 and 2019–2020 years. NINTN, traditional no irrigation with no topdressing N; EFINTN, every furrow irrigation with no topdressing N; AFINTN, alternative furrow irrigation with no topdressing N; EFITN, every furrow irrigation with topdressing N; AFITN, alternative furrow irrigation with topdressing N. Bars indicated standard deviation. Different letters within a cultivar indicate significant differences between groups ($p < 0.05$).

### 3.4. Grain Yield, Yield Components and Harvest Index

As shown in Table 6, both the FI methods and TN patterns significantly affected the yield and yield components of winter wheat, with different effects between years. In 2019–2020, the differences in grains per spike, 1000–grain weight, grain yield and harvest index among treatments were greater than those in 2018–2019. Compared with EFI treatment within the same TN pattern, AFI treatment did not affect spike numbers but significantly improved the grains per spike, 1000–grain weight and grain yield by 5.6%, 4.0% and 10.2%, respectively, averaged across the two years. In 2018–2019, there was no significant difference in harvest index (HI) between the AFI and EFI under the TN pattern. However, there was a significant increase in HI (6.1%) under AFI compared with EFI in 2019–2020. Compared with NTN treatment within the same FI method, TN treatment significantly improved the grain yield of winter wheat by 14.7–22.1% via significantly increasing the grains per spike (6.7–11.2%), 1000–grain weight (4.1–6.7%) and harvest index (4.7–5.2%). There was a different dose–response relationship for the grain yield with different FI methods under the TN pattern. The response appeared to be more positive (the grain yield increased greater) under EFI than AFI in both years.

**Table 5.** Aboveground dry matter accumulation and translocation of winter wheat affected by different treatments in the 2018–2019 and 2019–2020 years.

| Year | Treatment | DAA | PRT | POA | DAM | CRPDA |
|------|-----------|-----|-----|-----|-----|-------|
| | NINTN | 8301 c | 2872 a | 1374 d | 9675 e | 32.3 d |
| | EFINTN | 9926 a | 2973 a | 2454 c | 12,381 d | 45.2 c |
| 2018–2019 | AFINTN | 9640 b | 2128 bc | 3996 b | 13,636 c | 65.3 b |
| | EFITN | 10,095 a | 2335 b | 4133 b | 14,228 b | 63.9 b |
| | AFITN | 9831 ab | 1892 c | 5036 a | 14,867 a | 72.7 a |
| | NINTN | 7401 d | 1405 c | 1719 d | 9120 e | 55.2 d |
| | EFINTN | 9641 bc | 2517 a | 2107 c | 11,482 d | 45.6 c |
| 2019–2020 | AFINTN | 9411 c | 1989 b | 3219 b | 12,630 c | 61.8 b |
| | EFITN | 10,083 a | 2319 a | 3382 b | 13,465 b | 59.3 bc |
| | AFITN | 9761 b | 1868 b | 4207 a | 13,968 a | 69.2 a |
| | | | *F*-value | | | |
| Year (Y) | | 227.8 ** | 51.7 ** | 88.1 ** | 282.5 ** | 4.8 * |
| Treatment (T) | | 201.6 ** | 25.4 ** | 480.4 ** | 1315.9 ** | 97.7 ** |
| Y × T | | 7.9 ** | 21.9 ** | 19.4 ** | 2.5 ns | 23.2 ** |

Note: DAA, dry matter accumulation at anthesis stage; PRT, pre–anthesis dry matter translocation; POA, post–anthesis dry matter accumulation; DAM, dry matter accumulation at maturity stage; CRPOA, contribution rate of post–anthesis dry matter accumulation to grain. NINTN, traditional no irrigation with no topdressing N; EFINTN, every furrow irrigation with no topdressing N; AFINTN, alternative furrow irrigation with no topdressing N; EFITN, every furrow irrigation with topdressing N; AFITN, alternative furrow irrigation with topdressing N. Means in a column followed by the different lowercase letters within a year are significantly different at $p < 0.05$. * Significant at $p < 0.05$; ** significant at $p < 0.01$; ns, not significant.

**Table 6.** Grain yield, yield components and harvest index of winter wheat affected by different treatments in the 2018–2019 and 2019–2020 years.

| Year | Treatment | Efficient Spikes | Grains Per Spike | 1000–Grain Weight | Grain Yield | Harvest Index |
|------|-----------|------------------|------------------|-------------------|-------------|---------------|
| | NINTN | 475.9 d | 28.1 d | 42.6 c | 4853 e | 43.9 c |
| | EFINTN | 584.3 bc | 29.4 c | 42.5 c | 6202 d | 43.8 c |
| 2018–2019 | AFINTN | 568.8 c | 32.4 b | 44.6 b | 6999 c | 44.9 b |
| | EFITN | 601.4 ab | 32.3 b | 44.7 b | 7392 b | 45.5 a |
| | AFITN | 608.1 a | 33.5 a | 45.7 a | 7918 a | 46.6 a |
| | NINTN | 585.4 c | 20.8 e | 34.5 e | 3571 e | 34.2 e |
| | EFINTN | 640.9 b | 25.5 d | 37.5 d | 5209 d | 39.7 d |
| 2019–2020 | AFINTN | 661.4 ab | 26.8 c | 39.6 c | 5953 c | 41.2 c |
| | EFITN | 658.6 ab | 28.7 b | 40.6 b | 6515 b | 42.3 b |
| | AFITN | 661.5 a | 29.5 a | 41.9 a | 6943 a | 43.5 a |
| | | | *F*-value | | | |
| Year (Y) | | 284.2 ** | 2070.8 ** | 2284.4 ** | 625.5 ** | 14.8 ** |
| Treatment (T) | | 75.0 ** | 541.4 ** | 294.5 ** | 759.7 ** | 2.8 ns |
| Y × T | | 6.8 ** | 43.2 ** | 47.3 ** | 3.1 * | 30.4 ** |

Note: NINTN, traditional no irrigation with no topdressing N; EFINTN, every furrow irrigation with no topdressing N; AFINTN, alternative furrow irrigation with no topdressing N; EFITN, every furrow irrigation with topdressing N; AFITN, alternative furrow irrigation with topdressing N. Means in a column followed by the different lowercase letters within a year are significantly different at $p < 0.05$. * Significant at $p < 0.05$; ** significant at $p < 0.01$; ns, not significant.

### 3.5. Water Consumption (WC) and Water Use Efficiency (WUE)

The WC and WUE were strongly influenced by the treatments in both years (Table 7). Overall, under the NTN pattern, FI increased the $WC_{ba}$ by 71.6% (19.0 mm), $WC_{am}$ by 65.4% (24.1 mm) and ET by 18.0% (55.5 mm) compared with NINTN. In addition, compared with EFI treatment within the same TN pattern, AFI increased $WC_{am}$ (7.2−21.2%) in both years but did not significantly affect the $WC_{ba}$ and ET (except those under the TN pattern in the 2018–2019 year). Compared with NTN treatment, although the increase was not significant on the $WC_{ba}$ under NTN in the 2019–2020 year, TN significantly increased the $WC_{ba}$, $WC_{am}$ and ET by 9.8%, 9.5% and 3.0%, respectively, averaged across the two years. This indicated the positive effect of TN at jointing on water uptake by wheat in dryland.

The WUE was increased significantly under the application of EFI, AFI and TN techniques, with the highest WUE observed under AFITN treatment in both years (Table 7). Moreover, AFI tended to increase the WUE by 9.4% compared with EFI, and TN tended to increase the WUE by 15.2% compared with NTN. This result was similar to the relationship of the grain yield between AFI and EFI, as well as TN and NTN (Table 6).

**Table 7.** Water consumption (WC) and water use efficiency (WUE) of winter wheat affected by different treatments in the 2018–2019 and 2019–2020 years.

| Treatment | WC$_{ba}$ | | WC$_{am}$ | | ET | | WUE | |
|---|---|---|---|---|---|---|---|---|
| | 2018–2019 | 2019–2020 | 2018–2019 | 2019–2020 | 2018–2019 | 2019–2020 | 2018–2019 | 2019–2020 |
| NINTN | 28.8 c | 24.9 c | 42.1 e | 33.5 d | 264 d | 366 c | 18.4 e | 9.8 e |
| EFINTN | 45.7 b | 45.5 b | 57.2 d | 57.1 c | 316 c | 423 b | 19.6 d | 12.3 d |
| AFINTN | 47.0 b | 45.3 b | 69.3 b | 63.9 b | 320 bc | 423 b | 21.9 c | 14.1 c |
| EFITN | 43.4 b | 52.5 a | 61.6 c | 65.7 b | 322 b | 438 a | 23.0 b | 14.9 b |
| AFITN | 52.5 a | 53.1 a | 72.7 a | 70.4 a | 330 a | 437 a | 24.0 a | 15.9 a |
| | | | | *F*-value | | | | |
| Year (Y) | 14.9 ** | | 14.2 ** | | 35,874.0 ** | | 3371.8 ** | |
| Treatment (T) | 28.1 ** | | 252.2 ** | | 2102.2 ** | | 269.6 ** | |
| Y × T | 3.4 ns | | 10.3 ** | | 10.3 ** | | 2.7 ns | |

Note: WC$_{ba}$, water consumption from booting to anthesis; WC$_{am}$, water consumption from anthesis to maturity; ET, evapotranspiration over the whole winter wheat growing season. NINTN, traditional no irrigation with no topdressing N; EFINTN, every furrow irrigation with no topdressing N; AFINTN, alternative furrow irrigation with no topdressing N; EFITN, every furrow irrigation with topdressing N; AFITN, alternative furrow irrigation with topdressing N. Means in a column followed by the different lowercase letters within a year are significantly different at $p < 0.05$. ** significant at $p < 0.01$; ns, not significant.

### 3.6. Economic Benefits

As shown in Table 8, the economic benefits and benefit/input rate varied significantly among treatments (except the AFINTN and EFITN in the 2018–2019 growing season and the AFINTN and EFITN in the 2019–2020 growing season), with the highest value obtained under AFITN treatment in both years. The economic benefits under AFITN were 12.3%, 19.7%, 50.2% and 145.7% higher than those under EFITN, AFINTN, EFINTN and NINTN, respectively. The benefit/input rate under AFITN was 15.0%, 12.6%, 44.9% and 114.5% higher than that under AFINTN, EFITN, EFINTN and NINTN, respectively. Compared with EFI treatment within the same TN pattern, AFI treatment significantly improved the benefits and benefit/input rate by 12.3–26.0% and 15.0–28.7%, respectively, averaged across the two years. Furthermore, compared with the NTN treatment within the same FI method, TN treatment significantly improved the benefits and benefit/input rate by 20.2–34.9% and 12.6–26.0%, respectively. These results indicated that AFI and TN at jointing could significantly increase economic benefit and the benefit/input rate of winter wheat in drylands. However, as TN treatment increased the input of N fertilizer, the benefit/input rate under EFITN decreased by 4.2% in 2018–2019 compared with that under AFINTN.

### 3.7. Contributions of Different Factors (FI, C$_{fi}$; AFI, C$_{afi}$; TN, C$_{tn}$; and Interaction between AFI and TN, C$_{at}$) to the Winter Wheat Grain Yield and WUE

The analysis of the contributions of different factors (Table 9) showed that AFI combined with TN greatly increased the winter wheat grain yield and WUE. Four factors were responsible for this increase: (i) the FI factor (C$_{fi}$), (ii) the AFI factor (C$_{afi}$), (iii) the TN factor (C$_{tn}$) and the interaction between AFI and TN (C$_{at}$). We found that FI made the highest contribution among these four factors, which caused an increase of 2146–2382 kg ha$^{-1}$ for yield and 3.5–4.3 kg ha$^{-1}$ mm$^{-1}$ for WUE under the AFI method and 1349–1638 kg ha$^{-1}$ for yield and 1.3–2.5 kg ha$^{-1}$ mm$^{-1}$ for WUE under EFI method. The contribution to the grain yield and WUE of the TN factor was stronger than that of the AFI factor in both years, and the effects were greater under NTN than TN, as well as greater under EFI than AFI, indicating that it becomes difficult to improve the grain yield and WUE of winter

wheat under NTRFS through AFI and TN when the productivity and efficiency have been improved by EFI. Thus, a negative $C_{at}$ was observed for grain yield and WUE in both years.

**Table 8.** Economic benefit, and benefit/cost rate of winter wheat affected by different treatments in the 2018–2019 and 2019–2020 years.

| Treatment | Economic Benefit (USD ha$^{-1}$) | | Benefit/Input Rate | |
|---|---|---|---|---|
| | **2018–2019** | **2019–2020** | **2018–2019** | **2019–2020** |
| NINTN | 988 d | 509 e | 1.18 e | 0.61 d |
| EFINTN | 1408 c | 1040 d | 1.53 d | 1.12 c |
| AFINTN | 1729 b | 1343 c | 1.92 b | 1.49 b |
| EFITN | 1799 b | 1477 b | 1.84 c | 1.50 b |
| AFITN | 2018 a | 1660 a | 2.11 a | 1.73 a |
| | *F*-value | | | |
| Year (Y) | 644.1 ** | | 645.2 ** | |
| Treatment (T) | 613.8 ** | | 457.4 ** | |
| Y × T | 2.7 ns | | 5.9 * | |

Note: NINTN, traditional no irrigation with no topdressing N; EFINTN, every furrow irrigation with no top-dressing N; AFINTN, alternative furrow irrigation with no topdressing N; EFITN, every furrow irrigation with topdressing N; AFITN, alternative furrow irrigation with topdressing N. Means in a column followed by the different lowercase letters within a year are significantly different at $p < 0.05$. * Significant at $p < 0.05$; ** significant at $p < 0.01$; ns, not significant.

**Table 9.** Contributions of different factors (FI, $C_{fi}$; AFI, $C_{afi}$; TN, $C_{tn}$; and interaction between AFI and TN, $C_{at}$) to the winter wheat grain yield and WUE under different treatments in the 2018–2019 and 2019–2020 years.

| Years | Index | $C_0$ | $C_{fi}$ | | | $C_{afi}$ | | $C_{tn}$ | | $C_{at}$ |
|---|---|---|---|---|---|---|---|---|---|---|
| | | | NTN | EFI | AFI | NTN | TN | EFI | AFI | |
| 2018–2019 | Yield | 4853 | 1748 | 1349 | 2146 | 797 | 526 | 1190 | 919 | −393 |
| | WUE | 18.0 | 2.4 | 1.3 | 3.5 | 2.3 | 1.1 | 3.3 | 2.1 | −1.0 |
| 2019–2020 | Yield | 3571 | 2010 | 1638 | 2382 | 744 | 428 | 1306 | 990 | −562 |
| | WUE | 9.8 | 3.4 | 2.5 | 4.3 | 1.8 | 1.0 | 2.6 | 1.8 | −0.8 |

Note: $C_0$, yield (WUE) due to the baseline was equal to the grain yield (WUE) under NINTN treatment. $C_{fi}$, contribution of FI factor to Yield (WUE) was based on a comparison of the grain yield (WUE) under the EFINTN, AFINTN and NINTN treatments. $C_{afi}$, contribution of AFI factor to yield (WUE) was based on comparisons of the grain yield (WUE) under the AFI relative to the EFI under the NTN and TN treatments. $C_{tn}$, contribution of TN factor to yield (WUE) was based on comparisons of the grain yield (WUE) under the TN to the NTN under EFI and AFI treatments. $C_{at}$, contribution of AFI and TN factor interaction to yield (WUE), was based on the residual contribution to the total contribution amount (EFINTN/AFINTN/EFITN/AFITN grain yield– NINTN grain yield (WUE)) after subtracting a single AFI factor ($C_{afi}$) and a single TN factor ($C_{tn}$).

## 4. Discussion

The present study further confirmed the common perception that alternative furrow irrigation and topdressing N at jointing improve the grain yield and water use of winter wheat in semi–humid drought–prone areas. This study also found that alternative furrow irrigation combined with topdressing N (AFITN) performed much better than other combinations in terms of grain yield, leaf *Pn*, dry matter accumulation, WUE and economic benefit. Thus, this study provided new academic insights into how to manipulate the FI method and TN pattern to increase the benefits of irrigational water and N fertilizer in drought–prone areas with one–off irrigation.

### 4.1. Alternative Furrow Irrigation Combined with Topdressing Nitrogen at Jointing Optimized Soil Water

Water is deficient in dryland wheat production systems, and thus it is important to improve the water supply to wheat growth stages in order to ensure that an adequate yield is obtained. Previous studies concluded irrigation could significantly increase soil moisture and grain yield in semi–arid [13] and semi–humid drought–prone [32] areas of China. In the present study, the average SWS under the four FI treatments was 11.2%, 7.2% and 1.2% greater at booting, anthesis and maturity stages than that under NINTN

treatment, indicating that furrow irrigation (FI) with 75 mm at jointing increased the SWS at first, which then slowed down gradually along with wheat growth. The SWS under AFI treatment was increased at booting and anthesis stages but was decreased at maturity (especially in the 80–160 cm soil layer) (Figure 3), compared with that under EFI treatment. These results indicated that AFI of 75 mm at jointing would improve water supply from booting to anthesis and promote the $WC_{am}$ (Table 7). The reasons underlying this phenomenon may be related to the variation of the evaporation rate and water capture abilities between AFI and EFI.

Under EFI conditions, irrigation water accumulates mainly within the shallow soil layer. However, the limited irrigated water centralized into 50% furrows caused by AFI and the 2–fold water amount for one irrigation furrow under AFI made it easy for irrigation water to infiltrate into the deeper soil layers. Thus, the loss of irrigation water via evaporation was reduced under the AFI. Under this condition, the greater water supply in the deeper soil layer under AFI promoted the root development and the growth of aboveground parts of winter wheat [33,34], resulting in more water uptake by plant growth [35]. These results demonstrated that the AFI technique improved the availability of water to plants by efficiently centralizing the limited irrigational water into partly furrows compared with EFI. Some previous results are in accordance with our findings. For instance, Liu et al. [36] demonstrated that one–off irrigation at jointing (75 mm) significantly improved temporal and spatial root growth, made it easier to utilize soil water and soil N and created ideal circumstances for higher wheat production in Henan Province, China. Anapalli et al. [37] found that AFI had a higher daily ET (average of 5.8 mm $d^{-1}$ under the AFI vs. 5.6 mm $d^{-1}$ in the EFI treatment) and significantly increased the grain yield compared with EFI. These results indicate that one–off alternative furrow irrigation at jointing with 75 mm is suitable for winter wheat under the NTRFS system in the study area. Supplemental irrigation of 75 mm (half in winter and half at jointing) under the RF system was also reported as the best practice for significantly improving soil water storage, reducing evapotranspiration and increasing crop water productivity in semi–arid regions [38].

Additionally, we also found that the soil moisture under TN was less than that under NTN in both AFI and EFI methods, and the difference expanded gradually along with wheat growth. These were mainly because TN significantly increased the absorption of soil water by wheat compared with NTN; thus, there was a decrease in soil water from the booting to maturity stages (Table 4, Figure 3). The results were similar to a previous study by Zhang et al., who found that the application of N fertilizer could enhance crop drought resistance through increasing water uptake by the plant in dryland farming [4].

*4.2. Alternative Furrow Irrigation Combined with Topdressing Nitrogen at Jointing Increased Leaf SPAD and Pn and Improved Aboveground Dry Matter Accumulation*

Photosynthesis is the most essential process affecting crop production and can be easily inhibited by water stress [11,39]. In drought–prone regions of China, wheat production was limited by poor photosynthetic characteristics due to water shortages that lead to water deficiency during the reproductive growth stages [11,13,40]. Several studies generally recognized that increasing SWC improved *P*n, which contributes to increased crop production [11,13,41]. In dryland areas, previous research has shown that the AFI can significantly reduce water loss via leaf transpiration but will not significantly reduce the *P*n [10,42]. In the present study, AFI increased leaf SPAD (2.0–7.6%) and *P*n (1.5–7.5%) by alleviating the water stress at the anthesis stage and promoted water uptake by wheat compared with the EFI within the same TN pattern. Meanwhile, TN treatment increased the SPAD (3.2–10.7%) and *P*n (5.6–15.0%) by optimizing the N nutrition of winter wheat compared with NTN treatment. Thus, AFITN modified soil water status, resulting in a considerable improvement in SPAD and *P*n of leaves in wheat after anthesis. AFI had significantly positive effects on soil water storage at the booting and anthesis stages (Figure 3), while TN showed significantly positive effects on water consumption from booting to anthesis and from anthesis to maturity (Table 7). These positive effects under AFI and TN

explained the increased wheat SPAD and *P*n observed under AFITN. Previous studies also indicated that proper water and N management efficiently increased chlorophyll content and *P*n in dryland [11,38], and many studies have shown that leaf SPAD and *P*n in wheat increased under AFI and TN [13].

Winter wheat production, linked to the dry matter accumulation and the translocation of pre–anthesis assimilates [11,43], is very susceptible to drought stress during the reproductive growth stage [44]. Application of an efficient water use technology would improve the characteristic of aboveground dry matter accumulation and translocation in winter wheat [11,32]. In the present study, the DAM and POA were significantly improved with the application of FI, AFI and TN techniques (Table 5), which were conducive to building a good matter basis for yield formation. However, the translocation of pre–anthesis assimilates (PRT) was declined when these techniques were applied. This may be due to the senescence and degradation of the inner structure of the plant and the fact that stored matter mobilization is accelerated under drought, which promoted the re–translocation of the reserves accumulated before anthesis [40]. We also found that the increase in POA was 40.6%(AFI) −46.4% (TN) when AFI and TN were applied, and the two techniques notably interacted to affect the regulation of crop growth.

### 4.3. Alternative Furrow Irrigation Combined with Topdressing Nitrogen at Jointing Increased Grain Yield, WUE and Economic Benefit

High grain yields constitute the most important objective for wheat production in China [45]. Previous studies have suggested that AFI and TN can increase the grain yield and WUE of wheat [22,46,47]. A farm–scale field experiment in 2017 and 2019 in the Lower Mississippi Delta region of the United States showed that AFI with ~60% of the irrigations applied in EFI treatment significantly increased corn yield by 5.2% compared with EFI [37]. Leininger et al. [24] reported that AFI system enhanced lateral water suction from a non–irrigated furrow, which helped maintain yields in peanut compared with the FI system. Jia et al. [46] reported that, under the same amount of irrigation with 150 mm, AFI enhanced winter wheat yield via the significantly increased spike numbers compared with flood irrigation. In the present study, AFI indeed significantly increased wheat grain yield by increasing the harvest index, grains per spike and 1000–grain weight compared with EFI (Table 6). This may be ascribed to the increased SWS and the optimized soil water distribution under AFI treatments (Table 3, Figure 3), which had a significant positive influence on soil nutrient supply capacity, and then evapotranspiration and yield formation [48]. These results are in agreement with previous research in the semi–arid area of Kenya [49] and semi–humid area of China [32]. There was no significant difference in spike numbers between AFI and EFI, which indicated that the soil water supply with 75 mm irrigation at jointing was sufficient for the efficient spike establishment (mainly before the booting stage). TN along with irrigation is widely used in wheat production to ensure the N demand for plant growth and thus increases the grain yield [36]. We also found that TN treatment significantly improved the grain yield of winter wheat by significantly increasing the grains per spike, 1000–grain weight and harvest index compared with NTN treatments within the same FI method. Furthermore, TN had a significant synergistic effect with AFI on grain yield and yield components (Table 6). Finally, AFITN treatment significantly increased winter wheat yield in comparison with other treatments. This was mainly because AFI and TN techniques enhanced crop reproductive allocation due to the improved soil hydrothermal condition (Figure 3), the increased leaf *P*n (Figure 4), dry matter assimilates (Table 5) and yield components (Table 6).

Promoting WUE is one of the main objectives for sustainable agriculture, which is achieved by increasing crop yield and reducing ET, which comprises soil evaporation and plant transpiration [50]. Barideh et al. [51] concluded that the highest WUE was observed in the AFI method. Sarker et al. [52] showed that AFI had the highest WUE, owing to less water use, and produced better yield. In our study, the WUE of winter wheat increased as FI was applied, which was consistent with the previous studies [17,19,51,52].

Moreover, the WUE under AFI was greater than that under EFI within the same TN pattern. Furthermore, the WUE under TN was also higher than that under NTN within the same FI method. These are mainly because both AFI–induced yield and TN–induced yield were increased (Table 6). Previous studies have demonstrated that increasing the ratio of plant transpiration/evaporation was beneficial for saving water and thus increased crop WUE [50]. In the present study, AFI helped the irrigational water flow down to the subsoil and reduce the evaporation and led to greater water uptake by the plant, thereby increasing the transpiration/evaporation ratio. In addition, TN promoted wheat growth and increased the $WC_{am}$ mainly via water uptake by the plant. The increased ET (mainly by plant uptake) promoted the winter wheat growth and grain yield under AFI and TN, and the interaction was investigated between the FI method and TN pattern (Tables 6 and 7). These may be the main reasons why AFITN increased WUE.

Further analysis showed that cultivation management (including FI, AFI and TN) contributed to winter yield and WUE to a great extent, but it was interesting that FI made the highest contribution among the four factors comprising FI, AFI, TN and the interaction of AFI and TN. The contribution of AFI was greater under NTN than that under TN, and the contribution of TN was greater under EFI than that under AFI. In addition, the contribution of cultivation management to grain yield or WUE was determined by the amount of precipitation and air temperature. Notably, the grain yield and WUE in 2018–2019 were higher than those in 2019–2020, regardless of the irrigation methods and N patterns, owing to the higher annual rainfalls and 3–day high temperature (max temperature beyond 40 °C) at the anthesis stage in 2019–2020. This implies that other environmental factors such as high air temperature had a negative influence on wheat yield by reducing grains per spike and the 1000–grain weight in dryland (Table 6). However, the increase in AFI and TN on yield and WUE was greater in 2019–2020 than that in 2018–2019, indicating that both the AFI and TN could reduce the negative influence of environmental stress on wheat production, thus stabilizing the grain yield and WUE of winter wheat in dryland.

Economic benefit and benefit/input rate are important socio–economic factors when evaluating cultivation management methods and determining the adoption of dryland farming practices by farmers [13]. In the present study, although the inputs of expenditure and labor under FI and TN were raised, this could be offset by increased crop yield; thus, the economic benefit under FI and TN was significantly higher than that under NI and NTN. Furthermore, compared with EFI, AFI treatment significantly increased the economic benefit and benefit/cost rate due to the same expenditure and the increase in yield. Therefore, AFITN treatment achieved the highest economic benefit by 1660–2018 USD ha$^{-1}$, and the benefit/cost rate by 1.73–2.11. Compared with NINTN, EFINTN, AFINTN and EFITN, AFITN caused an increase in economic benefit of 145.7%, 50.2%, 19.7% and 12.3%, respectively, and an increase in the benefit/cost rate of 114.5%, 44.9%, 12.6% and 15.0%, respectively. This increase was mainly because of the increased grain yield in both growing seasons (Table 6). Ali et al. [38] also found that compared with no irrigation treatment, irrigation with 75 mm under the RF system increased the economic benefit and the benefit/input rate due to the significantly increased yield of winter wheat.

## 5. Conclusions

In the present study, it was found that furrow irrigation (FI) at jointing had a significantly positive effect on yield formation and water use of winter wheat in the no–till ridge furrow planting system. Alternative furrow irrigation (AFI) at jointing increased the SWS at the booting and anthesis stages, thereby enhancing the leaf $Pn$ and dry matter production, and ultimately led to the increase in the grain yield and WUE. Topdressing N (TN) at jointing along with irrigation increased the water absorption by winter wheat, and thus the grain yield and WUE were increased significantly under TN. However, the regulative effect of AFI was greater under NTN than that under TN, and the regulative effect of TN was greater under EFI than that under AFI, indicating there was a synergistic effect between the FI method and the TN pattern. Compared with the other treatments,

AFITN lead to the greatest grain yield, WUE and economic benefit with an increase of 6.6–94.4%, 4.3–6.6.2% and 12.3–145.5%, respectively. Therefore, AFITN can be a promising strategy for improving winter wheat yield and WUE in a no–till ridge furrow planting system in semi–humid drought–prone areas. Further research is needed to clarify that whether the amount and scheme of AFI and TN should be optimized for other dryland wheat production systems under different soil properties, i.e., soil texture, filed capacity, wilting point and electrical conductivity.

**Author Contributions:** J.W.: Performed the experiments; Investigation; Data curation; and Writing—Original draft and review and editing. H.G.: Performed data analysis and Data searching. Z.W., Y.L. and G.F.: Designed the experiments; Acquired Funding; and Revised the manuscript. M.H. and G.L.: Designed the experiments; Acquired Funding; Revised the manuscript; and Writing—Review and editing. All authors have read and agreed to the published version of the manuscript.

**Funding:** This study was financially supported by the National Key Research and Development Program of China (under grant no. 2016YFD0300404 and no. 2018YFD0300707) and the Key Subject Group Construction Project of Henan (under grant no. 17100001).

**Data Availability Statement:** This study includes all supporting data, which can be obtained from the corresponding authors upon request.

**Acknowledgments:** The author would like to thank the reviewers for their valuable comments and suggestions for this work.

**Conflicts of Interest:** The authors declare no conflict of interest.

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
