# Peer review of "Alternative Furrow Irrigation Combined with Topdressing Nitrogen at Jointing Help Yield Formation and Water Use of Winter Wheat under No-Till Ridge Furrow Planting System in Semi-Humid Drought-Prone Areas of China"

_agronomy, doi:10.3390/agronomy13051390_

Round 1

Reviewer 1 Report

Alternative furrow irrigation combined with topdressing nitro-2 gen at jointing help yield formation and water use of winter 3 wheat under no-till ridge furrow planting system in semi-humid drought-prone areas of China

This study is interesting topic that related the water use and nitrogen efficiency of the wheat crop, however, a major revision should be made before publication; also, extensive English proofreading is needed because of quite a number of grammatical errors and inappropriate words and expressions. Specifically, the main problems and suggestions are listed as follows.

1. Abstract

It’s necessary to present the innovative concepts or significant contributions of the study to arouse the readers’ interests. The present form of abstract could be improved, try to consolidate the abstract following the study purpose, methods and results in a readily readable manner, also it is essential to figure out the main conclusion of the study.

-The abstract is general and not figure out the innovation point of the paper, please rewrite it and mention the summary of the material section.

2. Introduction

The introduction part is too long and more useless information is there, please rewrite it to be consice and clear. Try to mention the reserach problem and how the research can solve it. What is the innovation point of this study.

3. Material and Methods

- How the authors calculated the amount of water consumed 289 over the whole winter wheat year?

- How much the irrigation amount every irrigation time?

- The section of ‘2.3.6. Contributions of different factors to the grain yield and WUE’ put it in table will be clear for reader.

- What does it mean ‘NINTN’ , 1/2EFINTN grain yield (WUE) + 297 1/2AFINTN grain yield (WUE)? It is very confusing to understand the treatments, please make it clear to east got the meaning.

- In figure 1, add the key of Monthly maximum/minimum air temperature beside precipitation.

Try to explain more the figure 2 to be understandable for readers.

- How the authors measure the Leaf SPAD and net photosynthesis rate and Aboveground dry matter accumulation?

- Use heatmap to figure out the results of the figure 3 instead of the current figure.

- The authors didn’t mention any information about the nitrogen use efficiency and runoff values which are more related and useful for the current study.

-It’s better to compare all examined parameters in the three stages to figure out each case what the difference is.

-Also, it is better to measure the gray water footprint during the season of wheat.

- What is the application uniformity of the Alternative furrow irrigation? In comparison with the sprinkler irrigation system, the Alternative furrow irrigation is lower than sprinkler irrigation, thus why the authors applied Alternative furrow irrigation?

- what is the water storage in different distance on the furrow from the beginning and the end of the furrow? What’s the total length of the furrow?

- How the authors calculate the irrigation amount of each treatment in table 2 ?

-The authors said the irrigation amount was 75 mm both in AFI and EFI treatments, is it the gross value or net value? And what the plant irrigation requirements?

Author Response

This study is interesting topic that related the water use and nitrogen efficiency of the wheat crop, however, a major revision should be made before publication; also, extensive English is needed because of quite a number of grammatical errors and inappropriate words and expressions. Specifically, the main problems and suggestions are listed as follows.

Response: Thank you for the positive comments on our research work, and we have revised the manuscript as you suggested. Also, the English proofreading of the present manuscript has been improved according to your suggestion.

The English was checking by Lipeng Wu, the PhD in food science from Zhejiang University. As the first author, he has published 5 peer-reviewed international research articles including Medicinal Research Reviews (IF, 12.94), Food Chemistry (two articles, IF, 9.23), Journal of Food Composition and Analysis (IF, 4.52) and Molecules (IF, 4.92). He also served as a reviwer for some journals such as Food Chemistry (IF, 9.23), Journal of Soil Science and Plant Nutrition (IF, 3.61), Phytotherapy Research (IF, 6.38), and Antioxidants (IF, 7.67).

The responds to your comments are as following:

  1. Abstract

It’s necessary to present the innovative concepts or significant contributions of the study to arouse the readers’ interests. The present form of abstract could be improved, try to consolidate the abstract following the study purpose, methods and results in a readily readable manner, also it is essential to figure out the main conclusion of the study.

Response: We revised the description of the background in the abstract of the present study for arousing the innovation point, and modified it as the style of purpose, methods, results and main conclusion. More detail please see our revised manuscript.

-The abstract is general and not figure out the innovation point of the paper, please rewrite it and mention the summary of the material section.

Response: We prepared the abstract as a general style. To summarized the material section, we revised this part as “In 2018–2020, a field experiment with two furrow irrigation (FI) practices (EFI and AFI: every furrow irrigation and alternative furrow irrigation with 75 mm at jointing stage, respectively) as main treatment, and two topdressing N (TN) patterns (NTN and TN: 0 and 60 kg ha-1along with irrigation, respectively) as secondary treatment, and a traditional planting practice with no irrigation and no topdressing N (NINTN) as the control, to investigate the effects of soil water, leaf chlorophyll relative content (SPAD) and net photosynthetic rate (Pn), aboveground dry matter assimilates, grain yield, and water use efficiency (WUE)”. We also figure out the innovation point as “This study provided new insights into improving wheat productivity in drought-prone areas where one-off irrigation was guaranteed”

  1. Introduction

The introduction part is too long and more useless information is there, please rewrite it to be concise and clear. Try to mention the research problem and how the research can solve it. What is the innovation point of this study.

Response: We simplified the introduction part as possible as we can. More detail please see our revised manuscript.

  1. Material and Methods

How the authors calculated the amount of water consumed 289 over the whole winter wheat year?

Response: Sorry for careless. The WC is the amount of water consumed over the whole winter wheat growing season, not over the whole year. And the WC was changed as ET, which is more specialized for explaining the water consumed over the whole winter wheat growing season.

- How much the irrigation amounts every irrigation time?

Response: In this study, the irrigation only conducted one time, and the irrigation amount was 75 mm based on the whole plot surface.

- The section of ‘2.3.6. Contributions of different factors to the grain yield and WUE’ put it in table will be clear for reader.

Response: We revised this section as your suggestion in Table 3. More detail please see our revised manuscript.

- What does it mean ‘NINTN’, 1/2EFINTN grain yield (WUE) + 1/2AFINTN grain yield (WUE)? It is very confusing to understand the treatments, please make it clear to east got the meaning.

Response: NINTN, EFINTN, AFINTN, EFITN, AFITN mean the yield and WUE value under these treatments. For making it more clear and easily got the meaning, we revised the style, for example, changed “1/2EFINTN grain yield (WUE)” to “1/2 grain yield (WUE) under EFINTN”

- In figure 1, add the key of Monthly maximum/minimum air temperature beside precipitation.

Response: Sorry for careless. The legend is not fully displayed. And we add it as your suggestion and revised the Fig1 (from September to June) as the suggestion of Reviewer#2.

Try to explain more the figure 2 to be understandable for readers.

Response: Thanks for your suggestion, we add some explanation in figure 2. More detail please see our revised manuscript.

How the authors measure the Leaf SPAD and net photosynthesis rate and Aboveground dry matter accumulation?

Response: The measurement progress and method of Leaf SPAD and net photosynthesis rate was described in 2.3.2; and the aboveground dry matter accumulation (kg ha–1) was calculated from the summed dry matter accumulation by each organ in 2.3.3. We revised the relative description for more acute and clear. More detail please see our revised manuscript.

Use heatmap to figure out the results of the figure 3 instead of the current figure.

Response: Sorry for the no revising, because we can not figure out the data of five treatments into a heatmap, and if change, 30 heatmaps is need in figure 3. Meanwhile, many studies relative to SWS are showed by line figure.

The authors didn’t mention any information about the nitrogen use efficiency and runoff values which are more related and useful for the current study. It’s better to compare all examined parameters in the three stages to figure out each case what the difference is. Also, it is better to measure the gray water footprint during the season of wheat.

Response: As your mention, nitrogen use efficiency, N runoff values and water footprint are related and useful for the current study. However, in this paper, we mainly focus on the yield formation and water use, and the contents include soil water storage at three growth stages, leaf SPAD and Pn at four stages, dry matter accumulation and translocation, grain yield and its components, and water consumption and WUE, and we also mentioned the contribution of different factors to yield and WUE. Generally, it is enough for a paper. Furthermore, the present manuscript is over 11000 words. Therefore, we are sorry for not adding the results of N use efficiency, and nitrate leaching, water foot print.

  As your suggestion, we compared the examined parameters in different stages more detail when the different is significant to figure out what the difference is. More detail please see our revised manuscript.

What is the application uniformity of the Alternative furrow irrigation? In comparison with the sprinkler irrigation system, the Alternative furrow irrigation is lower than sprinkler irrigation, thus why the authors applied Alternative furrow irrigation?

Response: Alternative furrow irrigation (AFI) is one of water-saving techniques, which not only having the advantages of traditional every furrow irrigation (EFI) method such as: direct and retaining water to the plant root zone in the planting furrow, water-saving, low cost, less complicated, easy implementation and convenient for household use, but also having the advantages: save irrigation water, make the irrigation water infiltrates into the deeper soil layer due to concentrating limited irrigation water into partial irrigation furrows, control transpiration, reduce plant redundancy growth, promote water and nutrient uptake by crop, and finally increased yield and efficiency of wheat. Moreover, in the sub-humid drouth prone area, the irrigation water is limited, and these advantages of AFI indicated that AFI is one of efficient techniques for using the limited irrigation water to gain higher yield and efficiency in wheat. These maybe why we applied AFI.

Furthermore, in production practice, we found that the uniformity of irrigation water under AFI perform well. First, the irrigation water was adopted just one furrow apart, and the interval was only 34 cm. The wheats in the non-irrigation furrow could uptake water from the irrigated furrow by root, and there was no significant difference in field performance of the aboveground part. Second, the flood irrigation is applied according local farmer practice and it is a traditional method, thus AFI concentrate the limited irrigation water into partial furrows is prone to increase the uniformity of water in different position. In comparison with the sprinkler irrigation system, the AFI is easy implementation and convenient, and less depend on the equipment and less expenditure. Furthermore, sometimes, the uniformity of sprinkler irrigation is not guaranteed due to the uneven spraying of equipment and windy weather, especially when the irrigation is not enough.

What is the water storage in different distance on the furrow from the beginning and the end of the furrow? What’s the total length of the furrow?

Response: As usually, the total length of the furrow is around 60-80 m for farmers, in the present study, in order to avoid the difference of water storage in different distance on the furrow from the beginning and the end of the furrow, the total length of the furrow is 20 m. And there indeed was no difference in different position according to the field performance of wheat.

- How the authors calculate the irrigation amount of each treatment in table 2 ?

Response: The irrigation amount was calculated based on plot surface area and controlled by mechanical water meter reading (accuracy 0.01m3, and the working pressure of outlet valve was 0.10–0.12MPa). In detail, the irrigation volume in each plot was 9.18 m3 and the irrigated area in one furrow was 2.8 m2 (20 m × 0.14 m); Under EFI treatments, the total irrigated area in each plot was 50.4 m2 (20× 2.52 m), and the irrigation volume in one irrigated furrow was 0.51 m3; under AFI treatments, the irrigated area in each plot was 25.2 m2 (20× 1.26 m), and the irrigated volumes in one irrigated furrow was 1.02 m3.

-The authors said the irrigation amount was 75 mm both in AFI and EFI treatments, is it the gross value or net value? And what the plant irrigation requirements?

Response: In the present study, the irrigation amount is net value, which was calculated based on the whole plot surface and controlled by mechanical water meter reading.

Reviewer 2 Report

Please try to reformulate in case of title, you must be more concise.

The experiment was very well organized and conducted, there are very important data. It would have been more appropriate if the research has focused on the differences between EFI and EFI.

-in figure 1 please present the climatic data  from September to June, what is specific to wheat growing season

- in material and methods section- please redefine the control variant (NINTN), this treatment is not zero nitrogen, these is a basic fertilizer application and TN is additional fertilizer on vegetative stage

- please indicate in material and methods section and also in table the control

- please try to simplify/if possible abbreviations used (for instance CRPODMATG, NINTN and others)

- please move some references from discussion part to introduction part if you don t compare your results with theirs (line 513, 515, 517)

-line 590- please correct TD into TN

-in the Discussion section please insist on the superior values of yield and yield components in AFI treatment and if you have some data regarding the roots development please add.

In my opinion the English language is ok.

Author Response

The experiment was very well organized and conducted, there are very important data. It would have been more appropriate if the research has focused on the differences between EFI and EFI.

Response: Thank you for your positive comments on our research work. In the present study, we intend to explore the effects of AFI and TN technique on NTRFS wheat based on one-off irrigation in sub-humid drought-prone area, thus we focused on the differences between AFI and EFI as well as the differences between TN and NTN. As your suggestion, we strengthened the analysis on the differences between AFI and EFI.

We have revised the manuscript as you suggested, and the responds are as following:

-in figure 1 please present the climatic data from September to June, what is specific to wheat growing season

Response: As your suggestion, we revised figure 1 and present the climatic data from September to June.

- in material and methods section- please redefine the control variant (NINTN), this treatment is not zero nitrogen, these is a basic fertilizer application and TN is additional fertilizer on vegetative stage

Response: In the present study, these was a same basic fertilizer application for all treatments, and the difference is the different water and N managements at jointing. The NINTN treatment was set based on the local farmer practice, in which, there was no irrigation and no topdressing N at jointing, thus we named it as NINTN. Thus, we think NTNTN is accurate and reasonable. Please understand that no modification was made for NINTN.

- please indicate in material and methods section and also in table the control.

Response: In the present study, we applied a two-factor experimental treatment and a control. Also, the NINTN was used as the control. Thus, there are three controls, NINTN to other treatments, EFI to AFI, NTN to TN. In order to express more clearly, we indicated the NINTN as control in material and methods section and also in Table 2 as your suggestion.

- please try to simplify/if possible abbreviations used (for instance CRPODMATG, NINTN and others).

Response: We change the CRPODMATG to CRPOA, as well as DMAA to DAA, PRDMT to PRT, PODMA to POA, DMAM to DAM.

Sorry, we do not change the abbreviations of NINTN, the reason is that the other treatments are named FITN or FINTN, as a control, the name of NINTN maybe more clear than other abbreviation.

- please move some references from discussion part to introduction part if you don t compare your results with theirs (line 513, 515, 517)

Response: These references are in accordance with our results, so we compare our results with these references and not move them.

-line 590- please correct TD into TN.

Response: Sorry for the carelessness, and we have correct TD into TN.

-in the Discussion section please insist on the superior values of yield and yield components in AFI treatment and if you have some data regarding the roots development please add.

Response: First, as your suggestion, we insist on the superior values of yield and yield components affected by AFI. Second, in the present study, we did not measure the root data. However, many literatures have been demonstrated that AFI is beneficial to root development, and we speculate the reason of yield and yield components in AFI treatment maybe the roots developments. The greater soil water consumption from booting to maturity in AFI may explain root development to some extent.

Reviewer 3 Report

Dear authors

Please find comments on the manuscript “Alternative furrow irrigation with topdressing nitrogen at jointing help yield formation and water use of winter wheat under no-till ridge furrow planting system in semi-humid drought-prone areas of China” submitted to Agronomy Journal.

Overall, I consider the text is good, however, some aspects must be improved.

In fact, a major aspect that called my attention is that making some simple calculations based on the measures for the furrows and ridges around 60% of the surface is lost, I mean, only 40% of one hectare will be effectively used to grow winter wheat, how this affects the income of the farmers? Also, the cost analysis must be provided considering the implementation of the furrows, does the increase in yield really pay for the soil management? no analysis was performed in soil management to improve the slope when furrows are made. This must be considered if furrows will be established in the long term, soil leveling should be considered to increase irrigation efficiency and decrease soil erosion.

Abstract

Introduction

Line 48: state the scientific name of winter wheat.

Line 49: 75% of winter wheat or 75% of all the wheat? Explain

First paragraph: explain why winter wheat is a major cereal crop, used mainly as a human food source, and is harvested in areas where water is not enough. How do you explain this contradiction?

Line 57: to increase rain WUE or increased? Must improve the English language across the whole text.

Line 79: I would suggest removing the sentence “furrow irrigation is a state-of-art strategy to address such problems”. In fact, furrow irrigation is one of the less developed irrigation techniques, in other arid areas of the world, companies are creating water pools to accumulate water and use a pivot system with over 90% efficiency, way higher than furrow that in the best case could reach 50%.

Lines 81-86: Furrow irrigation is one of the less efficient irrigation systems. Seems the selection of references here was totally biased.

 Materials and Methods

Soil properties: MUST include electrical conductivity and bulk density values. MUST include the physical properties of soil i.e. soil texture analysis for the analyzed profile (i.e. 0-20 and 20-40 cm). In addition, MUST provide an analysis of Field capacity and wilting point water retention. This is basic information that must be provided to understand the water holding capacity of the area of study and relate it with any potential water balance to be performed.

Line 199: it says “and the space of the two wheats was 20 cm in wide and 14 cm 199 in narrow, with the average of 17 cm.” I do not understand this sentence, please improve.

Line 212: why measure soil moisture from 0 to 200 cm? explain

MUST provide information related to crop coefficient and evapotranspiration. So far has not been possible to understand how you measured the water consumption, roughly you can estimate the gross water layer applied. In addition, must provide information on how the 75 mm for irrigation was determined.

I would suggest English for the section defining WUE, as far as I remember WUE is related to the amount of water applied as a gross amount of water, not to the water consumption, in fact, again evapotranspiration was not defined or any other water balance to estimate the consumption of the water by the crop.

Provide a summary table showing information related to descriptive statistics on all the measured variables. SPAD for example can have variations up to 35% in the same leaf even measured 1 or 2 millimeters from the first measure. This information is necessary to better understand de variability of the data, including the coefficient of variation among other descriptive statistics.

Conclusions

Please, check the values provided in lines 616-617.

Also, must improve, which seems a summary of the results.

References

Check references, sometimes you are using periods for the abbreviations and sometimes you are not.

I would suggest checking English writing, first pages were difficult to read in some sections.

Author Response

Please find comments on the manuscript “Alternative furrow irrigation with topdressing nitrogen at jointing help yield formation and water use of winter wheat under no-till ridge furrow planting system in semi-humid drought-prone areas of China” submitted to Agronomy Journal.

Overall, I consider the text is good, however, some aspects must be improved.

Response: Thank you for your positive comments on the text of our manuscript and the useful suggestion. We revised the manuscript as you suggested or explain why not, and the responds are as following:

In fact, a major aspect that called my attention is that making some simple calculations based on the measures for the furrows and ridges around 60% of the surface is lost, I mean, only 40% of one hectare will be effectively used to grow winter wheat, how this affects the income of the farmers? Also, the cost analysis must be provided considering the implementation of the furrows, does the increase in yield really pay for the soil management? no analysis was performed in soil management to improve the slope when furrows are made. This must be considered if furrows will be established in the long term, soil leveling should be considered to increase irrigation efficiency and decrease soil erosion.

Response: In the present study, the ride and furrow prepared by a no tillage fertilizer seeder (2BMQF-6/12A, Luoyang Xinle Machinery Co., Ltd), which can simultaneously conduct the operation of furrowing, ridging, fertilizing, sowing and repressing. There was no special consumption for madding ridge and furrow. Owing to there was no tillage and one-off field operation in this practice, the cost is low than that in the conventional tillage practice in farmer (plough, rotary, made furrow, etc.).

Here, we must explain, wheat is a crop grown in rows, as normal, the space of two-row wheat is 20 cm. In the NTRFS technique used in the present experiment, although there was a ridge (20cm), but the furrow is only 14cm, the average space of two-row wheat is 17cm, the row space is narrowed, and thus, there was no surface lost compared farmer practice.

AS your suggestion, we add the results of economic profit of different treatments.

Line 48: state the scientific name of winter wheat.

Response: As your suggestion, we state the scientific name (Triticum aestivuml) of winter wheat.

Line 49: 75% of winter wheat or 75% of all the wheat? Explain

Response: According to the reference, it is 75% of all the wheat, we revised the description.

First paragraph: explain why winter wheat is a major cereal crop, used mainly as a human food source, and is harvested in areas where water is not enough. How do you explain this contradiction?

Response: Although that “winter wheat is a major cereal crop, used mainly as a human food source and mainly planted in areas where water is not enough” is contradicted, but it is in line with the actual production and well known by agronomist. In fact, wheat (mainly of which is winter wheat) as a typical dry crop, accounts for 30% of the cultivated field in drylands (UNESCO, 2009). Thus, we think that there is no need to explain this contradiction. However, as your suggestion, we change the description as “Wheat is one of the stable food crops and feeds about 30% of the world population [1], and about 75% of all the wheat is produced from the dryland including arid, semi-arid, semi-humid drought-prone areas[2].

Line 57: to increase rain WUE or increased? Must improve the English language across the whole text.

Response: this is rain use efficiency, but not rain WUE, we also modified the language across the whole text.

Line 79: I would suggest removing the sentence “furrow irrigation is a state-of-art strategy to address such problems”. In fact, furrow irrigation is one of the less developed irrigation techniques, in other arid areas of the world, companies are creating water pools to accumulate water and use a pivot system with over 90% efficiency, way higher than furrow that in the best case could reach 50%.

Response: As your suggestion, we delete the sentence “furrow irrigation is a state-of-art strategy to address such problems”. In our study, we mention the FI as a developed irrigation technique its water-saving, low cost, less complicated, easy implementation and convenient for household use compared traditional flood irrigation method, and the best irrigation technique is AFI (the pivot irrigation system as you noted).

Lines 81-86: Furrow irrigation is one of the less efficient irrigation systems. Seems the selection of references here was totally biased.

Response: Here, we mention the advantages of furrow irrigation compared with traditional food irrigation, and the furrow irrigation include every furrow irrigation and alternative furrow irrigation. In fact, the production practice certificate that, with the same irrigation amount, the application of furrow irrigation within ride and furrow planting system produced higher water productivity and crop yield than conventional flood irrigation within the flat planting system or no irrigation system. Thus, we think the advantage of furrow irrigation exit, but it is less efficient than AFI. The results in the present study can also confirm this point.

 Materials and Methods

Soil properties: MUST include electrical conductivity and bulk density values. MUST include the physical properties of soil i.e. soil texture analysis for the analyzed profile (i.e. 0-20 and 20-40 cm). In addition, MUST provide an analysis of Field capacity and wilting point water retention. This is basic information that must be provided to understand the water holding capacity of the area of study and relate it with any potential water balance to be performed.

Response: We add the relative soil properties, such as and bulk density, and field capacity that we have measured at initial of the experimental.

However, we can not provide the electrical conductivity, soil texture and wilting point water retention. As you mentioned, this is important information for understanding the water holding capacity of the area of study and relate it with any potential water balance to be performed. Owing to in many previous agronomic articles with similar contents, the information of electrical conductivity, soil texture and wilting point water retention is not provided, thus we did not measure this index, so can not provide the information of these index, we are very sorry for these. 

Line 199: it says “and the space of the two wheats was 20 cm in wide and 14 cm 199 in narrow, with the average of 17 cm.” I do not understand this sentence, please improve.

Response: the sentence was improved as “Thus, the space of the two wheats in wide row was 20 cm in wide and while that in narrow row was 14 cm ”

Line 212: why measure soil moisture from 0 to 200 cm? explain

Response: as experience and previous relative literature, the soil moisture in wheat planting system mainly focuses on 0 to 200 cm. This is related to the fact that wheat roots are 2 m deep. So we measure soil moisture from 0 to 200 cm.

MUST provide information related to crop coefficient and evapotranspiration. So far has not been possible to understand how you measured the water consumption, roughly you can estimate the gross water layer applied. In addition, must provide information on how the 75 mm for irrigation was determined.

Response: We revised the information related to crop coefficient and evapotranspiration. In detail:

Soil water storage (SWS, mm) was calculated according to Zhao[15]:

Where Di is the soil bulk density (g cm3); Hi is the soil thickness of the i layer (cm); Wi is soil water content on a gravimetric basis (%); and n is the number of soil layers; i = 20, 40, 60, …, 200.

According to Zhang et al., (2020), the water consumption amount (WC, mm) was calculated as: WC = ΔSWS + P

Where ΔSWS (mm) is the difference in SWS between the beginning and the end of the wheat growth period, and P (mm) is the precipitation during the wheat growth period.

In the original manuscript, due the evapotranspiration is equal to WC during the wheat growing season and the describe as the relative literature. We calculated the WUE = WUE =Y ÷ WC, is not specification.

In the revised manuscript, we add the calculation of ET (evapotranspiration), and add the equation of ET, ET=WSs + P + U - R - F - WSm.

Also, we provide information on how the 75 mm for irrigation was determined in the field management section.

I would suggest English for the section defining WUE, as far as I remember WUE is related to the amount of water applied as a gross amount of water, not to the water consumption, in fact, again evapotranspiration was not defined or any other water balance to estimate the consumption of the water by the crop.

Response: As you mentioned, WUE is calculated as WUE =Y ÷ ET.

In the original manuscript, we calculated the WUE: WUE =Y ÷ WC, as the literature of Zhang (in order4), the reason is that evapotranspiration is equal to WC during the wheat growing season, and we calculated the WC from different growth stage of wheat. WUE =Y ÷ WC is easily misunderstood and not specification.

 In the revised manuscript, we add the calculation of ET (evapotranspiration), and add the formula of ET=WSs + P + U - R - F - WSm, and revised the calculation formula of WUE as WUE =Y ÷ ET.

Provide a summary table showing information related to descriptive statistics on all the measured variables. SPAD for example can have variations up to 35% in the same leaf even measured 1 or 2 millimeters from the first measure. This information is necessary to better understand de variability of the data, including the coefficient of variation among other descriptive statistics.

Response: As your mentioned, the measured index varied strongly. SPAD for example can have variations up to 35% in the same leaf even measured 1 or 2 millimeters from the first measure. So, in order to ensure the accuracy of data, we measured 10 times for SPAD and Pn. “Each time, ten leaves of the center rows of each plot were measured at the same leaf position avoided major veins” in the present study, we record the average as measurement value.

As your suggestion, we analyzed the variability of all the measured variables and try to provide the descriptive statistics. However, regardless of soil layers and stages, there were 23 variables in our study, if made in a table, It's too complicated. Therefore, we tend to provide the coefficient of variation as supplemental information if needed.

Please, check the values provided in lines 616-617. Also, must improve, which seems a summary of the results.

Response: Although the regulation effects of different treatments on most parameters is similar, the effects on the grain yield increases and WUE increases due to cultivation management were variable in our study, where they were determined significantly by the amount of rainfall and air temperature.

I would suggest checking English writing, first pages were difficult to read in some sections.

Response: We modified the English writing as your suggestion. And the English was checkin

Please find comments on the manuscript “Alternative furrow irrigation with topdressing nitrogen at jointing help yield formation and water use of winter wheat under no-till ridge furrow planting system in semi-humid drought-prone areas of China” submitted to Agronomy Journal.

Overall, I consider the text is good, however, some aspects must be improved.

Response: Thank you for your positive comments on the text of our manuscript and the useful suggestion. We revised the manuscript as you suggested or explain why not, and the responds are as following:

In fact, a major aspect that called my attention is that making some simple calculations based on the measures for the furrows and ridges around 60% of the surface is lost, I mean, only 40% of one hectare will be effectively used to grow winter wheat, how this affects the income of the farmers? Also, the cost analysis must be provided considering the implementation of the furrows, does the increase in yield really pay for the soil management? no analysis was performed in soil management to improve the slope when furrows are made. This must be considered if furrows will be established in the long term, soil leveling should be considered to increase irrigation efficiency and decrease soil erosion.

Response: In the present study, the ride and furrow prepared by a no tillage fertilizer seeder (2BMQF-6/12A, Luoyang Xinle Machinery Co., Ltd), which can simultaneously conduct the operation of furrowing, ridging, fertilizing, sowing and repressing. There was no special consumption for madding ridge and furrow. Owing to there was no tillage and one-off field operation in this practice, the cost is low than that in the conventional tillage practice in farmer (plough, rotary, made furrow, etc.).

Here, we must explain, wheat is a crop grown in rows, as normal, the space of two-row wheat is 20 cm. In the NTRFS technique used in the present experiment, although there was a ridge (20cm), but the furrow is only 14cm, the average space of two-row wheat is 17cm, the row space is narrowed, and thus, there was no surface lost compared farmer practice.

AS your suggestion, we add the results of economic profit of different treatments.

Line 48: state the scientific name of winter wheat.

Response: As your suggestion, we state the scientific name (Triticum aestivuml) of winter wheat.

Line 49: 75% of winter wheat or 75% of all the wheat? Explain

Response: According to the reference, it is 75% of all the wheat, we revised the description.

First paragraph: explain why winter wheat is a major cereal crop, used mainly as a human food source, and is harvested in areas where water is not enough. How do you explain this contradiction?

Response: Although that “winter wheat is a major cereal crop, used mainly as a human food source and mainly planted in areas where water is not enough” is contradicted, but it is in line with the actual production and well known by agronomist. In fact, wheat (mainly of which is winter wheat) as a typical dry crop, accounts for 30% of the cultivated field in drylands (UNESCO, 2009). Thus, we think that there is no need to explain this contradiction. However, as your suggestion, we change the description as “Wheat is one of the stable food crops and feeds about 30% of the world population [1], and about 75% of all the wheat is produced from the dryland including arid, semi-arid, semi-humid drought-prone areas[2].

Line 57: to increase rain WUE or increased? Must improve the English language across the whole text.

Response: this is rain use efficiency, but not rain WUE, we also modified the language across the whole text.

Line 79: I would suggest removing the sentence “furrow irrigation is a state-of-art strategy to address such problems”. In fact, furrow irrigation is one of the less developed irrigation techniques, in other arid areas of the world, companies are creating water pools to accumulate water and use a pivot system with over 90% efficiency, way higher than furrow that in the best case could reach 50%.

Response: As your suggestion, we delete the sentence “furrow irrigation is a state-of-art strategy to address such problems”. In our study, we mention the FI as a developed irrigation technique its water-saving, low cost, less complicated, easy implementation and convenient for household use compared traditional flood irrigation method, and the best irrigation technique is AFI (the pivot irrigation system as you noted).

Lines 81-86: Furrow irrigation is one of the less efficient irrigation systems. Seems the selection of references here was totally biased.

Response: Here, we mention the advantages of furrow irrigation compared with traditional food irrigation, and the furrow irrigation include every furrow irrigation and alternative furrow irrigation. In fact, the production practice certificate that, with the same irrigation amount, the application of furrow irrigation within ride and furrow planting system produced higher water productivity and crop yield than conventional flood irrigation within the flat planting system or no irrigation system. Thus, we think the advantage of furrow irrigation exit, but it is less efficient than AFI. The results in the present study can also confirm this point.

 Materials and Methods

Soil properties: MUST include electrical conductivity and bulk density values. MUST include the physical properties of soil i.e. soil texture analysis for the analyzed profile (i.e. 0-20 and 20-40 cm). In addition, MUST provide an analysis of Field capacity and wilting point water retention. This is basic information that must be provided to understand the water holding capacity of the area of study and relate it with any potential water balance to be performed.

Response: We add the relative soil properties, such as and bulk density, and field capacity that we have measured at initial of the experimental.

However, we can not provide the electrical conductivity, soil texture and wilting point water retention. As you mentioned, this is important information for understanding the water holding capacity of the area of study and relate it with any potential water balance to be performed. Owing to in many previous agronomic articles with similar contents, the information of electrical conductivity, soil texture and wilting point water retention is not provided, thus we did not measure this index, so can not provide the information of these index, we are very sorry for these. 

Line 199: it says “and the space of the two wheats was 20 cm in wide and 14 cm 199 in narrow, with the average of 17 cm.” I do not understand this sentence, please improve.

Response: the sentence was improved as “Thus, the space of the two wheats in wide row was 20 cm in wide and while that in narrow row was 14 cm ”

Line 212: why measure soil moisture from 0 to 200 cm? explain

Response: as experience and previous relative literature, the soil moisture in wheat planting system mainly focuses on 0 to 200 cm. This is related to the fact that wheat roots are 2 m deep. So we measure soil moisture from 0 to 200 cm.

MUST provide information related to crop coefficient and evapotranspiration. So far has not been possible to understand how you measured the water consumption, roughly you can estimate the gross water layer applied. In addition, must provide information on how the 75 mm for irrigation was determined.

Response: We revised the information related to crop coefficient and evapotranspiration. In detail:

Soil water storage (SWS, mm) was calculated according to Zhao[15]:

Where Di is the soil bulk density (g cm3); Hi is the soil thickness of the i layer (cm); Wi is soil water content on a gravimetric basis (%); and n is the number of soil layers; i = 20, 40, 60, …, 200.

According to Zhang et al., (2020), the water consumption amount (WC, mm) was calculated as: WC = ΔSWS + P

Where ΔSWS (mm) is the difference in SWS between the beginning and the end of the wheat growth period, and P (mm) is the precipitation during the wheat growth period.

In the original manuscript, due the evapotranspiration is equal to WC during the wheat growing season and the describe as the relative literature. We calculated the WUE = WUE =Y ÷ WC, is not specification.

In the revised manuscript, we add the calculation of ET (evapotranspiration), and add the equation of ET, ET=WSs + P + U - R - F - WSm.

Also, we provide information on how the 75 mm for irrigation was determined in the field management section.

I would suggest English for the section defining WUE, as far as I remember WUE is related to the amount of water applied as a gross amount of water, not to the water consumption, in fact, again evapotranspiration was not defined or any other water balance to estimate the consumption of the water by the crop.

Response: As you mentioned, WUE is calculated as WUE =Y ÷ ET.

In the original manuscript, we calculated the WUE: WUE =Y ÷ WC, as the literature of Zhang (in order4), the reason is that evapotranspiration is equal to WC during the wheat growing season, and we calculated the WC from different growth stage of wheat. WUE =Y ÷ WC is easily misunderstood and not specification.

 In the revised manuscript, we add the calculation of ET (evapotranspiration), and add the formula of ET=WSs + P + U - R - F - WSm, and revised the calculation formula of WUE as WUE =Y ÷ ET.

Provide a summary table showing information related to descriptive statistics on all the measured variables. SPAD for example can have variations up to 35% in the same leaf even measured 1 or 2 millimeters from the first measure. This information is necessary to better understand de variability of the data, including the coefficient of variation among other descriptive statistics.

Response: As your mentioned, the measured index varied strongly. SPAD for example can have variations up to 35% in the same leaf even measured 1 or 2 millimeters from the first measure. So, in order to ensure the accuracy of data, we measured 10 times for SPAD and Pn. “Each time, ten leaves of the center rows of each plot were measured at the same leaf position avoided major veins” in the present study, we record the average as measurement value.

As your suggestion, we analyzed the variability of all the measured variables and try to provide the descriptive statistics. However, regardless of soil layers and stages, there were 23 variables in our study, if made in a table, It's too complicated. Therefore, we tend to provide the coefficient of variation as supplemental information if needed.

Please, check the values provided in lines 616-617. Also, must improve, which seems a summary of the results.

Response: Although the regulation effects of different treatments on most parameters is similar, the effects on the grain yield increases and WUE increases due to cultivation management were variable in our study, where they were determined significantly by the amount of rainfall and air temperature.

I would suggest checking English writing, first pages were difficult to read in some sections.

Response: We modified the English writing as your suggestion. And the English was checking by Lipeng Wu, the PhD in food science from Zhejiang University. As the first author, he has published 5 peer-reviewed international research articles including Medicinal Research Reviews (IF, 12.94), Food Chemistry (two articles, IF, 9.23), Journal of Food Composition and Analysis (IF, 4.52) and Molecules (IF, 4.92). He also served as a reviwer for some journals such as Food Chemistry (IF, 9.23), Journal of Soil Science and Plant Nutrition (IF, 3.61), Phytotherapy Research (IF, 6.38), and Antioxidants (IF, 7.67).

g by Lipeng Wu, the PhD in food science from Zhejiang University. As the first author, he has published 5 peer-reviewed international research articles including Medicinal Research Reviews (IF, 12.94), Food Chemistry (two articles, IF, 9.23), Journal of Food Composition and Analysis (IF, 4.52) and Molecules (IF, 4.92). He also served as a reviwer for some journals such as Food Chemistry (IF, 9.23), Journal of Soil Science and Plant Nutrition (IF, 3.61), Phytotherapy Research (IF, 6.38), and Antioxidants (IF, 7.67).

Round 2

Reviewer 1 Report

I am agreeing with the current version of the manuscript which is significantly improved and ready for publication. However, it is still one part needs to revise and improve for final acceptance. The part is the economic section which is very useful for readers.  To reach the project cost of the current system, a feasibility study must be carried out to obtain the cost of the production of ton wheat per cubic meter of water. The cost should be in Dollar to be readable for all readers. The authors need to point out the fixed and running cost during the growing season.

Author Response

Dear Editors and Reviewers:

Thank you for your letter and for the comments concerning our manuscript entitled “Alternative furrow irrigation combined with topdressing nitrogen at jointing help yield formation and water use of winter wheat under no-till ridge furrow planting system in semi-humid drought-prone areas of China” (ID: agronomy-2367928). Those comments are all valuable and very helpful for revising and improving our paper, as well as the important guiding significance to our researches.

We have studied the referees’comments carefully and have made correction which we hope meet with approval. We also checked all references to ensure their relevant to the contents of the manuscript. Revised portion are marked up using the“Track Changes” function. The main corrections in the paper and the responds are as flowing:

Reviewer#1

I am agreeing with the current version of the manuscript which is significantly improved and ready for publication. However, it is still one part needs to revise and improve for final acceptance. The part is the economic section which is very useful for readers.  To reach the project cost of the current system, a feasibility study must be carried out to obtain the cost of the production of ton wheat per cubic meter of water. The cost should be in Dollar to be readable for all readers. The authors need to point out the fixed and running cost during the growing season.

Response: Thank you for the positive comments on our revise work, and we have revised the economic section of manuscript as you suggested.

 (I) to be readable for all readers, we changed the RMB to Dollar based on the average exchange rate between RMB and US dollar in 2018-2019 and 2019-2020 year;

 (II) The input value was fixed but not running due to the fixed field management and the fixed input of materials, labor and machinery application. Thus, the costs were same in the two growing seasons. However, in the study area and in China, the main cost of irrigation is fee of the used electricity for bombing water and labor for irrigation, which is about 30-40 RMB for 50 m3 (equal to the amount of 75mm for 667 m2). The cost of irrigation is 6.8%-9.4% of the total cost, but it brought an increase of output from grain yield by 28.0%-45.9%, and obtained the water production value of 12.2-14.8 RMB m-3 based on the output of per cubic meter of water. Therefore, this amount of irrigation in the present study is feasibility.

However, because the amount of irrigation was same in all treatments except NINTN, thus the difference of “production of ton wheat per cubic meter of water” is determined by the difference of grain yield. So, we did not present the cost of the production of ton wheat per cubic meter of water. We look forward to you can understand.

Reviewer#3

Dear authors, please provide information related to the monetary unit used in the RMB analysis (i.e. US dollars, Euro, etc) by the unit of surface.

Response: Thanks for your suggestion. We have modified the RMB to US dollars.

Please include in the conclusions that further research must include missing information that was not considered in this work, for example, wilting point and electrical conductivity.

Response: Thanks for your suggestion. We have modified the further research part as “Further research is needed to clarify that whether the amount and scheme of AFI and TN should be optimized for other dryland wheat production systems under different soil properties, i.e. soil texture, filed capacity, wilting point and electrical conductivity”.

Comments on the Quality of English Language: Suggesting a minor revision of English style, basically because it is notorious that a lot of work has been performed to improve the english style in the text. Usually after extensive use of tracking changes sometimes minor errors might be missed. Please read carefully before submission.

Response: To avoid the errors owing to the extensive tracking changes, we accept the Track Changes of the last uploaded manuscript, and read carefully to check errors.

Others

1: The data of economic benefit section were changed from RMB to Dollar, and thus the increases of AFI and TN were changed due to rounding of a value. We checked and revised the relative parts.

2.We modified the Figure 2 due to the big gap of the schematic diagram between the EFI and AFI methods.

Reviewer 3 Report

Dear authors, please provide information related to the monetary unit used in the RMB analysis (i.e. US dollars, Euro, etc) by the unit of surface.

Please include in the conclusions that further research must include missing information that was not considered in this work, for example, wilting point and electrical conductivity.

Suggesting a minor revision of english style, basically because it is notorious that a lot of work has been performed to improve the english style in the text. Usually after extensive use of tracking changes sometimes minor errors might be missed. Please read carefully before submission.

Author Response

Dear Editors and Reviewers:

Thank you for your letter and for the comments concerning our manuscript entitled “Alternative furrow irrigation combined with topdressing nitrogen at jointing help yield formation and water use of winter wheat under no-till ridge furrow planting system in semi-humid drought-prone areas of China” (ID: agronomy-2367928). Those comments are all valuable and very helpful for revising and improving our paper, as well as the important guiding significance to our researches.

We have studied the referees’comments carefully and have made correction which we hope meet with approval. We also checked all references to ensure their relevant to the contents of the manuscript. Revised portion are marked up using the“Track Changes” function. The main corrections in the paper and the responds are as flowing:

Dear authors, please provide information related to the monetary unit used in the RMB analysis (i.e. US dollars, Euro, etc) by the unit of surface.

Response: Thanks for your suggestion. We have modified the RMB to US dollars.

Please include in the conclusions that further research must include missing information that was not considered in this work, for example, wilting point and electrical conductivity.

Response: Thanks for your suggestion. We have modified the further research part as “Further research is needed to clarify that whether the amount and scheme of AFI and TN should be optimized for other dryland wheat production systems under different soil properties, i.e. soil texture, filed capacity, wilting point and electrical conductivity”.

Comments on the Quality of English Language: Suggesting a minor revision of English style, basically because it is notorious that a lot of work has been performed to improve the english style in the text. Usually after extensive use of tracking changes sometimes minor errors might be missed. Please read carefully before submission.

Response: To avoid the errors owing to the extensive tracking changes, we accept the Track Changes of the last uploaded manuscript, and read carefully to check errors.

Others

1: The data of economic benefit section were changed from RMB to Dollar, and thus the increases of AFI and TN were changed due to rounding of a value. We checked and revised the relative parts.

2.We modified the Figure 2 due to the big gap of the schematic diagram between the EFI and AFI methods.
